# Anonymized Histograms
# in Intermediate Privacy Models

**Badih Ghazi**     **Pritish Kamath**     **Ravi Kumar**     **Pasin Manurangsi**

Google Research

Mountain View, CA, US

badihghazi@gmail.com, pritish@alum.mit.edu,
ravi.k53@gmail.com, pasin@google.com

## Abstract

We study the problem of privately computing the *anonymized histogram* (a.k.a. *unattributed histogram*), which is defined as the histogram without item labels. Previous works have provided algorithms with $\ell_1$- and $\ell_2^2$-errors of $O_\varepsilon(\sqrt{n})$ in the central model of differential privacy (DP).

In this work, we provide an algorithm with a nearly matching error guarantee of $\widetilde{O}_\varepsilon(\sqrt{n})$ in the shuffle DP and pan-private models. Our algorithm is very simple: it just post-processes the discrete Laplace-noised histogram! Using this algorithm as a subroutine, we show applications in privately estimating symmetric properties of distributions such as entropy, support coverage, and support size.

## 1  Introduction

Computing histograms is among the most well-studied tasks in data analytics and machine learning. Suppose that there is a domain $[D] := \{1, \ldots, D\}$, where the $i$th user's input is $z_i \in [D]$. The *histogram* of the users' inputs $\{z_1, \ldots, z_n\}$ is defined as $\boldsymbol{h} := (h_1, \ldots, h_D)$ where $h_j := |\{i \in [n] \mid z_i = j\}|$, i.e., the number of users who contribute item $j \in [D]$. For many tasks, however, the different items themselves are not important and it instead suffices to consider the *anonymized histogram* (a.k.a. *unattributed histogram*) corresponding to $\boldsymbol{h}$, which is defined as $\boldsymbol{n_h} := (n^{(1)}, \ldots, n^{(D)})$ where $n^{(\ell)}$ denotes the $\ell$th largest element among the $h_j$'s. Whenever $\boldsymbol{h}$ is clear from context, we will skip the subscript and denote the anonymized histogram as simply $\boldsymbol{n}$.

Anonymized histograms have several applications including estimating symmetric properties of discrete distributions [11, 2, 16, 31], privately releasing the degree-distributions in social networks [34, 33, 40], and anonymizing password frequency lists [13, 14]. For more details, we refer the reader to [44] and the references therein.

In this work, we study private anonymized histograms. The notion of privacy we study is differential privacy (DP) [24, 23], which has emerged as a very popular notion of private data analysis leading to numerous practical deployments [27, 43, 30, 8, 21, 1, 36, 41].

Multiple works have studied the problem of computing private anonymized histogram, with the focus so far being on the central model of DP.[1] Moreover, two measures of error have been studied: $\ell_1$-error and $\ell_2^2$-error[2]. For the $\ell_2^2$-error case, Hay et al. [34] give an $\varepsilon$-DP algorithm with

---

[1]In the *central model* of DP, a trusted curator has access to the raw inputs of the users, and is supposed to output a DP estimate of the desired function, in this case, the anonymized histogram.

[2]Defined as $\|\boldsymbol{n} - \hat{\boldsymbol{n}}\|_1 = \sum_{j \in [D]} |n^{(j)} - \hat{n}^{(j)}|$ and $\|\boldsymbol{n} - \hat{\boldsymbol{n}}\|_2^2 = \sum_{j \in [D]} (n^{(j)} - \hat{n}^{(j)})^2$, where $\hat{\boldsymbol{n}}$ denotes the output anonymized histogram.

an expected error of $\widetilde{O}(\sqrt{n}/\varepsilon^2)^3$. As for the $\ell_1$-error, Blocki et al. [13] observed that the exponential mechanism yields an expected $\ell_1$-error of $O(\sqrt{n}/\varepsilon)$ since there are at most $\exp(O(\sqrt{n}))$ anonymized histograms in total [32]; recently, this bound was improved to $O(\sqrt{n\log(1/\varepsilon)}/\varepsilon)$ by Suresh [44]. On the lower bound front, Alda and Simon [5] proved a bound of $\Omega(\sqrt{n/\varepsilon})$ for the expected $\ell_1$-error; recently, this was improved to $\Omega(\sqrt{n\log(1/\varepsilon)/\varepsilon})$ by Manurangsi [37], matching the aforementioned upper bound of [44] to within a constant factor. The latter lower bound also applies to $(\varepsilon, \delta)$-DP algorithms for any sufficiently small $\delta$ (depending only on $\varepsilon$).

The anonymized histogram problem generalizes the COUNT-DISTINCT problem, which asks for the number of items $j$ such that $h_j > 0$. COUNT-DISTINCT can be easily solved in the central DP model by applying the discrete Laplace mechanism. In the (non-interactive) local DP setting, Chen et al. [17] proved a lower bound of $\Omega_\varepsilon(n)$, which means that one cannot asymptotically beat the trivial algorithm that always outputs zero. The strong lower bounds on the error incurred by protocols in the local setting generally motivate the study of intermediate models of privacy including the pan-private [25] and shuffle DP [12, 26, 18] models. In these models, it turns out that COUNT-DISTINCT can be solved to within $\widetilde{O}_\varepsilon(\sqrt{n})$-error while lower bounds of $\Omega_\varepsilon(\sqrt{n})$ are known [38, 9].

In this work, we show that, surprisingly, the anonymized histogram problem (which seems much harder than COUNT-DISTINCT) can in fact be solved with essentially the same asymptotic error of $\widetilde{O}_\varepsilon(\sqrt{n})$ as COUNT-DISTINCT, in both the pan-private and shuffle DP models. On the other hand, the aforementioned lower bound [17] for COUNT-DISTINCT also implies an $\Omega_\varepsilon(n)$ lower bound for the expected $\ell_1$-error of anonymized histogram in the more challenging local DP model. In other words, in the typical case where $\varepsilon$ is an absolute constant, it is impossible to achieve any asymptotic advantage over the trivial algorithm that always outputs the all-zeros histogram.

## 1.1 Our Results

A prominent approach for computing private histograms in the central DP model is to add discrete[4] Laplace noise to each histogram entry. We show that there is a post-processing algorithm that takes such a noised histogram and produces an accurate estimate of the anonymized histogram:

**Theorem 1** (Informal; Theorems 9 and 21). *There is an algorithm that takes in a noisy histogram, where an independent discrete Laplace noise of parameter $1/\varepsilon$ is added to each entry, and outputs an approximate anonymized histogram such that the expected $\ell_1$- and $\ell_2^2$-errors are[5] $\widetilde{O}_\varepsilon(\sqrt{n+D})$.*

Note that there is a dependency of $\sqrt{D}$ in the error bound in Theorem 1; when the domain size is large, this can dominate the $\sqrt{n}$ term. Fortunately, we show that this can be overcome by first randomly hashing into $B$ buckets before computing the histogram. By picking $B$ to be $O(n)$, we show that one can achieve an error that is $\widetilde{O}_\varepsilon(\sqrt{n})$ *without* any dependency on $D$:

**Theorem 2** (Informal; Corollary 13 and Theorem 22). *There is an algorithm that takes in a noisy hashed histogram, where an independent discrete Laplace noise of parameter $1/\varepsilon$ is added to each bucket after hashing, and outputs an approximate anonymized histogram such that the expected $\ell_1$-error is $\widetilde{O}_\varepsilon(\sqrt{n})$. A similar algorithm holds, which uses two noisy hashed histograms and achieves expected $\ell_2^2$-error of $\widetilde{O}_\varepsilon(\sqrt{n})$.*

Random hashing and computing discrete Laplace-noised histograms can be implemented in the pan-private[6] and shuffle DP settings [29, 10], where in the latter case we have to concede $\delta > 0$ in the privacy parameter. Thus, the theorem above yields:

**Corollary 3.** *For any $\varepsilon > 0, \delta \in (0, 1]$, there is an $\varepsilon$-DP algorithm for anonymized histogram in the pan-private model and an $(\varepsilon, \delta)$-DP algorithm in the shuffle DP model, with expected $\ell_1$- and $\ell_2^2$-errors of $\widetilde{O}_\varepsilon(\sqrt{n})$.*

---

[3] In fact, Hay et al. [34] proved a slightly stronger expected error bound of $O(d\log^3 n/\varepsilon^2)$, where $d$ denotes the number of unique values appearing in $\boldsymbol{n}$. Since there are at most $\sqrt{n}$ different values in the worst case, this guarantee yields an $O(\sqrt{n}\log^3 n/\varepsilon^2)$ bound for any instance.

[4] Our results can be adapted to continuous Laplace noise [24] with only a constant factor overhead in the error. We use discrete noise since it can be applied in the shuffle DP model (which is discrete in nature).

[5] $\widetilde{O}(f)$ denotes $O(f\log^c f)$ for some constant $c > 0$ and $O_\varepsilon(f)$ denotes $O(g(\varepsilon)f)$ for some function $g(\cdot)$.

[6] This is done by starting with $D$ i.i.d. discrete Laplace r.v.'s and incrementing each entry for the next item.

As an immediate application of the above, we get algorithms for estimating symmetric properties of distributions; a distribution property is said to be *symmetric* if it remains unchanged under relabeling of the domain symbols. For any (non-private) symmetric estimator with low sensitivity, we get a private estimator in the pan-private and shuffle DP models.

**Theorem 4** (Informal; Theorem 17). *For all $\varepsilon > 0, \delta \in (0,1]$, and distributions $\mathcal{D}$, for any symmetric distribution property $f$, and any symmetric estimator $\hat{f}$, there exists an $\varepsilon$-DP mechanism $\mathcal{M}$ in the pan-private model and an $(\varepsilon, \delta)$-DP mechanism $\mathcal{M}$ in the shuffle DP model, such that $\mathcal{M}$ outputs an $\alpha$-approximation to $f(\mathcal{D})$ with high probability. The sample complexity of the mechanism $\mathcal{M}$ is given as $C_{\hat{f}}(f, \alpha) + D_{\hat{f}}(\alpha, \varepsilon)$, where the first term is the non-private sample complexity of $\hat{f}$ and the second term depends on the sensitivity of $\hat{f}$.*

In particular, in Section 5, we apply the above result to estimate the Shannon entropy, support coverage, and support size of discrete distributions, which, to the best of our knowledge, is the first such sample complexity bound in the pan-private and shuffle DP models.

## 1.2 Overview of Techniques

We now describe the high-level ideas of our algorithms and proofs. For ease of exposition, we will occasionally be informal; all details are formalized in later sections. To describe our algorithm, we need definitions of prevalence and cumulative prevalence.

**Definition 5** (Prevalence and Cumulative Prevalence). The *prevalence* of a histogram $\boldsymbol{h}$ is defined as $\boldsymbol{\varphi}^{\boldsymbol{h}} := (\varphi_0^{\boldsymbol{h}}, \ldots, \varphi_n^{\boldsymbol{h}})$, where $\varphi_r^{\boldsymbol{h}} := |\{j \in [D] : h_j = r\}|$ is the number of entries with value $r$. The *cumulative prevalence* of a histogram $\boldsymbol{h}$ is defined as $\boldsymbol{\varphi}_{\geq}^{\boldsymbol{h}} := (\varphi_{\geq 1}^{\boldsymbol{h}}, \ldots, \varphi_{\geq n}^{\boldsymbol{h}})$, where $\varphi_{\geq r}^{\boldsymbol{h}} := |\{j \in [D] : h_j \geq r\}|$ is the number of entries with value at least $r$.

Prevalence and cumulative prevalence can be similarly defined for an anonymized histogram $\boldsymbol{n}$; note that $\boldsymbol{\varphi}^{\boldsymbol{h}} = \boldsymbol{\varphi}^{\boldsymbol{n}_h}$ and $\boldsymbol{\varphi}_{\geq}^{\boldsymbol{h}} = \boldsymbol{\varphi}_{\geq}^{\boldsymbol{n}_h}$. An important property of cumulative prevalence is that it preserves the $\ell_1$-distance.

**Observation 6.** For all anonymized histograms $\boldsymbol{n}, \hat{\boldsymbol{n}}$, it holds that $\|\boldsymbol{\varphi}_{\geq}^{\boldsymbol{n}} - \boldsymbol{\varphi}_{\geq}^{\hat{\boldsymbol{n}}}\|_1 = \|\boldsymbol{n} - \hat{\boldsymbol{n}}\|_1$.

We stress that, while cumulative prevalence has been used before in DP algorithms for computing anonymized histograms [44, 37], these algorithms require access to the true anonymized histogram first and therefore will only work in the central DP model.

**Algorithm for $\ell_1$-error.** For each $j \in [D]$, using the discrete Laplace-noised count, we produce an unbiased estimate for whether $h_j \geq r$ for each $r \in [n]$. Adding these up over all $j \in [D]$ gives an unbiased estimate for $\varphi_{\geq r}^{\boldsymbol{h}}$. We then "project" the estimated $\boldsymbol{\varphi}_{\geq}$ back so that it corresponds to a valid anonymized histogram. It can be seen that this last step can at most double the error.

While the algorithm described above is simple, it is unclear why it incurs an error of $\widetilde{O}_\varepsilon(\sqrt{n + D})$. The analysis turns out to be quite delicate. The key is that the unbiased estimator we use has variance that decreases exponentially with $|h_j - r|$. (In other words, the uncertainty is high only when $h_j$ is close to $r$.) Roughly speaking, this means that the total error is dominated by the error in the case where $h_j = r$. Suppose for simplicity that we only focus on this case. Since there are $\varphi_r^{\boldsymbol{h}}$ entries satisfying the condition, the expected $\ell_1$-error for $\boldsymbol{\varphi}_{\geq r}$ will be $\widetilde{O}_\varepsilon(\sqrt{\varphi_r^{\boldsymbol{h}}})$. Thus, in total, the $\ell_1$-error of the estimated cumulative prevalence is dominated by $\widetilde{O}_\varepsilon(\sum_{r \in [n]} \sqrt{\varphi_r^{\boldsymbol{h}}})$. We can now apply the Cauchy–Schwarz inequality to yield $\sum_{r \in [n]} \sqrt{\varphi_r^{\boldsymbol{h}}} \leq \sqrt{\sum_{r \in [n]} 1/r} \cdot \sqrt{\sum_{r \in [n]} r \cdot \varphi_r^{\boldsymbol{h}}} = \Theta(\sqrt{n \log n})$, where the last equality follows from the fact that $\sum_{r \in [n]} r \cdot \varphi_r^{\boldsymbol{h}}$ is simply the total counts in the histogram. This concludes our proof sketch. The full proof can be found in Section 3.

**Handling Large Domain Sizes.** When $D \gg n$, we randomly hash into $B = \widetilde{O}(n)$ buckets and compute the noisy "reduced" histogram on these $B$ buckets. We can use our approach above to compute the anonymized histogram on these $B$ buckets with $\ell_1$-error at most $\widetilde{O}_\varepsilon(\sqrt{n})$. While this is a reasonable approach, it does not yet give a good estimate for the original anonymized histogram: the reason is that there could be as many as $\widetilde{\Omega}(n)$ collisions due to hashing. To handle this, we

define a function that "inverts" the reduced anonymized histogram to the original anonymized histogram. We then show that (i) this inverse has constant "sensitivity" and (ii) the reduced histogram is concentrated around its mean with an $\ell_1$-deviation of $\widetilde{O}(\sqrt{n})$. Combining these two allows us to conclude that the inverse of the noisy anonymized histogram has expected $\ell_1$-error of $\widetilde{O}_\varepsilon(\sqrt{n})$ as desired. See Section 4 for details.

**Privately Computing Symmetric Properties of Distributions.** A large body of work starting with [2] has shown that plug-in estimators using the so-called profile maximum likelihood (PML) distribution achieve nearly optimal sample complexity for many symmetric distribution properties. At the core of these works is an important fact that many symmetric distribution properties have estimators (based on the anonymized histogram) with low sensitivity. Our algorithms then apply these estimators on our private anonymized histogram. The sensitivity of the estimator, together with the $\ell_1$-error bound we have shown for our anonymized histogram, immediately yield bounds on the errors of the estimators. See Section 5 for details.

**Algorithm for $\ell_2^2$-error.** Adapting the algorithm for $\ell_2^2$-error proceeds as follows: recall (e.g., by Hölder's inequality) that $\|\boldsymbol{n} - \hat{\boldsymbol{n}}\|_2^2 \leq \|\boldsymbol{n} - \hat{\boldsymbol{n}}\|_1 \cdot \|\boldsymbol{n} - \hat{\boldsymbol{n}}\|_\infty$. Notice also that the concentration of the noise implies that each discrete Laplace-noised count is within $O(\log D/\varepsilon)$ of the true value. Due to this, we may only change the last (i.e., "projection") step by adding an extra constraint that each entry of the output estimated anonymized histogram is within $O(\log D/\varepsilon)$ of the corresponding entry of the noisy histogram. This way we have ensured that $\|\boldsymbol{n} - \hat{\boldsymbol{n}}\|_\infty \leq O(\log D/\varepsilon)$. Combining this with our earlier bound on the expected $\ell_1$-error immediately yields the desired bound on the $\ell_2^2$-error. See Appendix B for details.

## 1.3 Other Related Work

The original paper on the pan-private model [25] also studies the problem of estimating $\varphi_r^{\boldsymbol{n}}$. However, their focus is on algorithms with small space complexity, and, if one were to sum up their error bounds for all $r$ directly, it would yield a trivial bound of $\Omega(n)$ on the total error in the anonymized histogram.

Several recent works have studied the testing/estimation/learning of symmetric and other properties of distributions under privacy constraints, mostly in the central DP model [3, 4, 6, 15, 20, 35] and some in the local DP model [22]. In particular, Acharya et al. [3] study privately computing symmetric distribution properties in the central model. Indeed, they also exploit the fact that the estimators have low sensitivity. However, in the central DP model, low sensitivity allows one to get a private estimate by adding Laplace noise directly to the non-private estimate, whereas we need to compute the estimator from our approximate anonymized histogram.

## 2 Differential Privacy

In this section, we review the basics of differential privacy (DP) and the shuffle DP and the pan-private models. Let $[n]$ denote the set $\{1, \ldots, n\}$ and let $\mathbf{1}[\cdot]$ denote the binary indicator function.

Two datasets $S = \{z_1, \ldots, z_n\}$ and $S' = \{z_1', \ldots, z_n'\}$ are said to be *neighboring*, denoted $S \sim S'$, if there is an index $i \in [n]$ such that $z_j = z_j'$ for all $j \in [n] \setminus \{i\}$. We recall the following definition [24, 23]:

**Definition 7** (Differential Privacy (DP)). Let $\varepsilon > 0$ and $\delta \in [0, 1]$. A randomized algorithm $\mathcal{M} : \mathcal{Z}^n \to \mathcal{R}$ is $(\varepsilon, \delta)$-*differentially private* $((\varepsilon, \delta)$-*DP*) if, for all $S \sim S'$ and all (measurable) outcomes $E \subseteq \mathcal{R}$, we have that $\Pr[\mathcal{M}(S) \in E] \leq e^\varepsilon \cdot \Pr[\mathcal{M}(S') \in E] + \delta$.

We denote $(\varepsilon, 0)$-DP as $\varepsilon$-DP or *pure*-DP. The case when $\delta > 0$ is referred to as *approximate*-DP.

In the *central DP* model, all the inputs are stored and processed by an analyzer and the privacy is enforced only on the output of the analyzer.

**Shuffle DP [12, 26, 18].** In the *shuffle DP* model, there are three algorithms, namely, a local randomizer $\mathcal{R}$, a shuffler $\mathcal{S}$, and an analyzer $\mathcal{A}$. Let $S = \{z_1, \ldots, z_n\}$ be the input dataset. The randomizer $\mathcal{R}$ takes $z \in S$ as input and outputs a multiset of messages. The shuffler $\mathcal{S}$ takes the

multiset of messages obtained from $\mathcal{R}$ applied to each $z \in S$ and permutes them randomly. The analyzer $\mathcal{A}$ takes this permuted multiset and computes the final output. The privacy in the shuffle model is enforced on the output of the shuffler $\mathcal{S}$, when a single input is changed.

**Pan-privacy [25].** In the pan-private model, there is an algorithm that takes in a data stream of unbounded length consisting of elements in the domain. It is required that the internal state of the algorithm after any number of steps satisfies $\varepsilon$-DP over the data stream prefix until that step.

We consider two data streams to be neighboring iff they differ by a single entry. This is sometimes referred to as the "event-level" or "record-level" setting [7], in contrast to the "user-level" setting that was the focus of the original work on pan-privacy [25]. We remark that the "user-level" setting is not appropriate for our notions of error as it allows for changing an entry of the histogram arbitrarily, meaning that we cannot achieve any non-trivial error guarantees. Furthermore, our definition above (which allows us to directly implement discrete Laplace-noised histogram) is different from the origin definition in [25], because the latter requires that revealing both the internal state and the final output *simultaneously* are DP. Nonetheless, our algorithm can be easily adapted for this more restricted model, but with a worse dependency on $\varepsilon$. We explain this in more detail in Appendix E.

**Histograms in Central DP.** For a distribution $D$, let $x \sim D$ denote that the random variable $x$ is chosen from $D$. For $p \in (0,1)$, the *discrete Laplace distribution* (aka symmetric geometric distribution), denoted by $\mathrm{DLap}(p)$, is the distribution supported on $\mathbb{Z}$ whose probability mass at $i \in \mathbb{Z}$ is $\frac{1-p}{1+p} \cdot p^{|i|}$.

We use the following well-known fact about central DP and histograms.

**Fact 8.** *The algorithm that adds $\mathrm{DLap}(e^{-\varepsilon/2})$ noise to each entry of a histogram is $\varepsilon$-DP in the central model.*

We refer to the output of this algorithm as the *discrete Laplace-noised* histogram.

For a histogram $\boldsymbol{h}$, we use $\boldsymbol{n_h}$ to denote the anonymized histogram corresponding to $\boldsymbol{h}$, often dropping the subscript whenever $\boldsymbol{h}$ is clear from context.

## 3 Post-Processing Noised Histogram

In this section we describe our main algorithm, which obtains an anonymized histogram by suitably post-processing a noised histogram. We first define the following function $f : \mathbb{Z} \to \mathbb{R}$, which is used in our post-processing method described in Algorithm 1.

$$f(m) = \begin{cases} 1 & \text{if } m > 0, \\ 1 + \frac{p}{(1-p)^2} & \text{if } m = 0, \\ -\frac{p}{(1-p)^2} & \text{if } m = -1, \\ 0 & \text{if } m < -1. \end{cases} \tag{1}$$

---

**Algorithm 1** Anonymized Histogram Estimator w.r.t. $\ell_1$ loss.

**Input:** Discrete Laplace-noised histogram $\boldsymbol{h'}$, i.e., $h'_j \sim h_j + \mathrm{DLap}(p)$

**for** $r \in [n]$ **do**
  $\hat{\varphi}_{\geq r} \leftarrow \sum_{j \in [D]} f(h'_j - r)$, where $f$ is defined in (1)
**return** $\hat{\boldsymbol{n}}$ that minimizes $\|\boldsymbol{\varphi}^{\hat{\boldsymbol{n}}}_{\geq} - \hat{\boldsymbol{\varphi}}_{\geq}\|_1$.

---

We give an efficient implementation of the above algorithm in Algorithm 5 (Appendix D.1), which runs in time $O(D + n \log n)$. The main idea is to compute $\hat{\boldsymbol{n}}$ that minimizes $\|\boldsymbol{\varphi}^{\hat{\boldsymbol{n}}}_{\geq} - \hat{\boldsymbol{\varphi}}_{\geq}\|_1$ using $\ell_1$-isotonic regression. We now state the main guarantee of the post-processing method.

**Theorem 9.** *For all histograms $\boldsymbol{h}$ with $\|\boldsymbol{h}\|_1 = n$, the estimate $\hat{\boldsymbol{n}}$ returned by Algorithm 1 satisfies*

$$\mathbb{E}[\|\hat{\boldsymbol{n}} - \boldsymbol{n}\|_1] \leq O\left(\sqrt{C_p(n+D)\log n}\right).$$

*where, $C_p := \frac{p^2}{(1-p)^5} + \frac{p}{1-p}$.*

From Fact 8, for $\varepsilon$-DP, we may set $p = e^{-\varepsilon/2}$. Theorem 9 then gives a bound of $O(\sqrt{(n+D)\log n}/\varepsilon^{2.5})$ on the expected $\ell_1$-error.

The crucial result needed in our analysis is the guarantee of the individual estimator $f$. We show that $f(h'_j - r)$ is an unbiased estimator of $\mathbf{1}[h_j \geq r]$ and furthermore its variance decreases (exponentially) as $|h_j - r|$ increases.

**Lemma 10.** *For all $h, r \in \mathbb{N} \cup \{0\}$, if $h' \sim h + \mathrm{DLap}(p)$, then it holds that*

$$\mathbb{E}[f(h'-r)] = \mathbf{1}[h \geq r], \quad and \tag{2}$$

$$\mathrm{Var}[f(h'-r)] \leq 4p^{|h-r|+1}\left(\tfrac{p}{(1-p)^3} + (1-p)\right). \tag{3}$$

*Proof.* Let $\tau := h - r$, $\tau' := h' - r$, and $x := p/(1-p)^2$. Consider $g : \mathbb{Z} \to \mathbb{R}$ defined by $g(m) := f(m) - 1/2$. Notice that (2) is equivalent to $\mathbb{E}[g(\tau')] = \mathbf{1}[\tau \geq 0] - 1/2$. Due to symmetry, it suffices to consider the case where $\tau \geq 0$. In this case, we have

$$\mathbb{E}[g(\tau')] = \mathbb{E}_{Z \sim \mathrm{DLap}(p)}[g(\tau + Z)] = \Pr[Z > -\tau] \cdot (1/2) + \Pr[Z \leq -\tau] \cdot \mathbb{E}[g(\tau + Z) \mid Z \leq -\tau].$$

The last term can be expanded as follows:

$$\begin{aligned}
&\mathbb{E}[g(\tau + Z) \mid Z \leq -\tau] \\
&= (1/2 + x) \cdot \Pr[Z = -\tau \mid Z \leq -\tau] - (1/2 + x) \cdot \Pr[Z = -\tau - 1 \mid Z \leq -\tau] \\
&\quad - (1/2) \cdot \Pr[Z < -\tau - 1 \mid Z \leq -\tau] \\
&= (1/2 + x) \cdot (1-p) - (1/2 + x) \cdot p(1-p) - (1/2) \cdot p^2 = 1/2.
\end{aligned}$$

Combining the two equations above, we arrive at $\mathbb{E}[g(\tau')] = \Pr[Z > -\tau] \cdot (1/2) + \Pr[Z \leq -\tau] \cdot (1/2) = 1/2$, thereby proving (2).

To prove (3), notice again that $\mathrm{Var}[f(\tau')] = \mathrm{Var}[g(\tau')]$. Again, due to symmetry, we may only consider the case $\tau \geq 0$. Here, we have

$$\begin{aligned}
\mathrm{Var}[g(\tau')] &= \mathbb{E}_{Z \sim \mathrm{DLap}(p)}[(g(\tau + Z) - 1/2)^2] \\
&= \Pr[Z \leq -\tau] \cdot \mathbb{E}[(g(\tau + Z) - 1/2)^2 \mid Z \leq -\tau] \\
&\leq p^\tau \cdot \mathbb{E}[(g(\tau + Z) - 1/2)^2 \mid Z \leq -\tau].
\end{aligned}$$

Similar to before, we can expand the last term as

$$\begin{aligned}
&\mathbb{E}[(g(\tau + Z) - 1/2)^2 \mid Z \leq -\tau] \\
&= x^2 \cdot \Pr[Z = -\tau \mid Z \leq -\tau] + (1+x)^2 \cdot \Pr[Z = -\tau - 1 \mid Z \leq -\tau] \\
&\quad + 1 \cdot \Pr[Z < -\tau - 1 \mid Z \leq -\tau] \\
&= x^2 \cdot (1-p) + (1+x)^2 \cdot p(1-p) + p^2 \\
&\leq p^2/(1-p)^3 + 2(1+x^2) \cdot p(1-p) + p^2 \\
&\leq 4p^2/(1-p)^3 + 2p(1-p).
\end{aligned}$$

Plugging this into the above, we get $\mathrm{Var}[g(\tau')] \leq 4p^{\tau+1}(p/(1-p)^3 + (1-p))$ as desired. $\qquad\square$

The following is an immediate consequence of Lemma 10, by summing over all $j \in [D]$.

**Observation 11.** *For all $r \in [n]$, and $\kappa := 4p\left(\tfrac{p}{(1-p)^3} + (1-p)\right)$ it holds that*

$$\mathbb{E}[\hat{\varphi}_{\geq r}] = \varphi_{\geq r}^{\boldsymbol{n}} \quad and \quad \mathrm{Var}[\hat{\varphi}_{\geq r}] \leq \sum_{\ell=0}^n \kappa \cdot p^{|\ell - r|} \cdot \varphi_\ell^{\boldsymbol{n}}.$$

*Proof of Theorem 9.* The expected $\ell_1$-error is bounded as

$$\begin{aligned}
\|\hat{\boldsymbol{n}} - \boldsymbol{n}\|_1 &= \|\boldsymbol{\varphi}_\geq^{\hat{\boldsymbol{n}}} - \boldsymbol{\varphi}_\geq^{\boldsymbol{n}}\|_1 \leq \|\boldsymbol{\varphi}_\geq^{\hat{\boldsymbol{n}}} - \hat{\boldsymbol{\varphi}}_\geq\|_1 + \|\hat{\boldsymbol{\varphi}}_\geq - \boldsymbol{\varphi}_\geq^{\boldsymbol{n}}\|_1 \leq 2 \cdot \|\boldsymbol{\varphi}_\geq^{\hat{\boldsymbol{n}}} - \hat{\boldsymbol{\varphi}}_\geq\|_1 \\
&\implies \|\hat{\boldsymbol{n}} - \boldsymbol{n}\|_1 \leq 2 \cdot \sum_{r \in [n]} |\varphi_{\geq r}^{\hat{\boldsymbol{n}}} - \hat{\varphi}_{\geq r}|, \tag{4}
\end{aligned}$$

where we use that $\|\varphi^{\hat{n}}_{\geq} - \hat{\varphi}_{\geq}\|_1 \leq \|\hat{\varphi}_{\geq} - \varphi^{n}_{\geq}\|_1$ by our choice of $\hat{n}$. From Observation 11, we have

$$\mathbb{E}\left[|\varphi^{\hat{n}}_{\geq r} - \hat{\varphi}_{\geq r}|\right] \leq \sqrt{\mathrm{Var}[\hat{\varphi}_{\geq r}]} \leq \sqrt{\kappa} \cdot \sqrt{\sum_{\ell=0}^{n} p^{|\ell-r|} \cdot \varphi^{n}_{\ell}}.$$

Combining this with (4), we have

$$\begin{aligned}
\mathbb{E}[\|\hat{n} - n\|_1] &\leq 2 \cdot \sum_{r \in [n]} \mathbb{E}\left[|\varphi^{\hat{n}}_{\geq r} - \hat{\varphi}_{\geq r}|\right] \\
&\leq 2\sqrt{\kappa} \cdot \left(\sum_{r \in [n]} \sqrt{\sum_{\ell=0}^{n} p^{|\ell-r|} \cdot \varphi^{n}_{\ell}}\right) \\
&\leq 2\sqrt{\kappa} \cdot \sqrt{\sum_{r \in [n]} \tfrac{1}{r}} \cdot \sqrt{\sum_{r \in [n]} r \cdot \left(\sum_{\ell=0}^{n} p^{|\ell-r|} \cdot \varphi^{n}_{\ell}\right)} \quad \text{(Cauchy–Schwarz)} \\
&\leq \sqrt{\kappa} \cdot O(\sqrt{\log n}) \cdot \sqrt{\sum_{\ell=0}^{n} \varphi^{n}_{\ell} \cdot \left(\sum_{r \in [n]} r \cdot p^{|\ell-r|}\right)} \\
&\leq \sqrt{\kappa} \cdot O(\sqrt{\log n}) \cdot \sqrt{\sum_{\ell=0}^{n} \varphi^{n}_{\ell} \cdot 2(\ell+1) \cdot \left(\sum_{t=0}^{\infty}(t+1) \cdot p^{t}\right)} \\
&= \sqrt{\kappa} \cdot O(\sqrt{\log n}) \cdot \sqrt{\sum_{\ell=0}^{n} \varphi^{n}_{\ell} \cdot 2(\ell+1) \cdot (1/(1-p)^2)} \\
&= \sqrt{\kappa/(1-p)^2} \cdot O(\sqrt{\log n}) \cdot \sqrt{\left(\sum_{\ell=0}^{n} \ell \cdot \varphi^{n}_{\ell}\right) + \left(\sum_{\ell=0}^{n} \varphi^{n}_{\ell}\right)} \\
&= \sqrt{\kappa/(1-p)^2} \cdot O(\sqrt{\log n}) \cdot \sqrt{n + D}. \qquad \qquad \square
\end{aligned}$$

## 4 Reducing Domain Size via Hashing

In this section we propose an algorithm to handle the case where $D \gg n$. The approach in this case is to hash the domain into something smaller. Let $B \in \mathbb{N}$ be the number of hash values. The distribution of the anonymized histogram produced after random hashing into $B$ buckets is equivalent to the following process:

- ▶ Let $n = (n^{(1)}, \ldots, n^{(D)})$ be the starting anonymized histogram.
- ▶ Pick a uniformly random hash function $H : [D] \to [B]$.
- ▶ Let $h^{\mathrm{red}} := (h^{\mathrm{red}}_1, \ldots, h^{\mathrm{red}}_B)$ denote the reduced histogram given by $h^{\mathrm{red}}_i = \sum_{j \in H^{-1}(i)} n^{(j)}$.
- ▶ Let $n^{\mathrm{red}}$ denote the corresponding anonymized histogram.

Let $\Gamma^B$ be the mapping from $n$ to $\mathbb{E}[\varphi^{n^{\mathrm{red}}}_{\geq}]$ where $n^{\mathrm{red}}$ is generated as above, and the expectation is over the choice of random hash functions $H$. With this notation, we present Algorithm 2.

---

**Algorithm 2** Anonymized Histogram Estimator w.r.t. $\ell_1$ loss, for large domains.

**Input:** Discrete Laplace-noised histogram $\tilde{h}^{\mathrm{red}}$, i.e., $\tilde{h}^{\mathrm{red}}_j \sim h^{\mathrm{red}}_j + \mathrm{DLap}(p)$

Compute an estimate $\hat{n}^{\mathrm{red}}$ of $n^{\mathrm{red}}$ using Algorithm 1
**return** $\hat{n}$ that minimizes $\|\Gamma^B(\hat{n}) - \varphi^{\hat{n}^{\mathrm{red}}}_{\geq}\|_1$, s.t. $\|\hat{n}\|_1 = n$

---

We give an efficient implementation of (a variant of) the above algorithm in Algorithm 7 (Appendix D.2), which runs in time $\widetilde{O}_\varepsilon(D + n \log n)$. The main result of this section is the following:

**Theorem 12.** *For all $B > 4n$ and histograms $h$ with $\|h\|_1 \leq n$, the estimate $\hat{n}$ returned by Algorithm 2 satisfies*

$$\begin{aligned}
\mathbb{E}[\|\hat{n} - n\|_1] &\leq O(\mathbb{E}[\|\hat{n}^{\mathrm{red}} - n^{\mathrm{red}}\|_1] + n \log n/\sqrt{B}) \\
&\leq O\left(\sqrt{C_p(n+B)\log n} + n \log n/\sqrt{B}\right) \quad \text{(using Theorem 9)}.
\end{aligned}$$

By setting $B = n\sqrt{\log n}$, we get the following corollary.

**Corollary 13.** *For all $0 < \varepsilon \leq O(1)$, Algorithm 2 for $p = e^{-\varepsilon/2}$ and $B = n\sqrt{\log n}$ is an $\varepsilon$-DP algorithm, and achieves an expected $\ell_1$-error of $E(n, \varepsilon) = O(\sqrt{n}(\log n)^{3/4}/\varepsilon^{2.5})$.*[7]

---

[7] By choosing $B = n\sqrt{\varepsilon^{2.5} \log n}$, the bound in Corollary 13 could be improved to $O(\sqrt{n}((\log n)^{3/4}/\varepsilon^{1.25} + (\log n)^{1/2}/\varepsilon^{2.5}))$ for $\varepsilon \geq \Omega(\log^{-0.4} n)$; we state the simpler bound for brevity.

We describe the main steps in the proof of Theorem 12.

**Lipschitzness of Inverse of $\Gamma^B$.** We start by showing that the "inverse" of $\Gamma^B$ is $O(1)$-Lipschitz:

**Lemma 14.** *[Proof in Appendix A.1] For $B > 4n$ and all anonymized histograms $\boldsymbol{n}, \boldsymbol{n}'$ with $\|\boldsymbol{n}\|_1, \|\boldsymbol{n}'\|_1 \leq n$,*

$$\|\Gamma^B(\boldsymbol{n}) - \Gamma^B(\boldsymbol{n}')\|_1 \geq \|\boldsymbol{n} - \boldsymbol{n}'\|_1/4 \,.$$

**Concentration of $n^{\mathrm{red}}$.** We now bound the expected $\ell_1$-distance between $\boldsymbol{\varphi}_{\geq}^{\boldsymbol{n}^{\mathrm{red}}}$ and its expectation $\Gamma^B(\boldsymbol{n})$.

**Lemma 15.** *Assume that $B \geq 2n$. Then, $\mathbb{E}[\|\boldsymbol{\varphi}_{\geq}^{\boldsymbol{n}^{\mathrm{red}}} - \Gamma^B(\boldsymbol{n})\|_1] \leq O(n \log n/\sqrt{B})$.*

*Proof.* We have

$$
\begin{aligned}
\mathbb{E}[\|\boldsymbol{\varphi}_{\geq}^{\boldsymbol{n}^{\mathrm{red}}} - \Gamma^B(\boldsymbol{n})\|_1] &= \sum_{r \in [n]} \mathbb{E}[|\varphi_{\geq r}^{\boldsymbol{n}^{\mathrm{red}}} - \Gamma^B(\boldsymbol{n})_r|] \\
&= \sum_{r \in [n]} \mathbb{E}[|\varphi_{\geq r}^{\boldsymbol{n}^{\mathrm{red}}} - \mathbb{E}[\varphi_{\geq r}^{\boldsymbol{n}^{\mathrm{red}}}]|] \\
&\leq \sum_{r \in [n]} \sqrt{\mathrm{Var}[\varphi_{\geq r}^{\boldsymbol{n}^{\mathrm{red}}}]}. \qquad (5)
\end{aligned}
$$

We next use the following bound on the variance.

**Lemma 16.** *[Proof in Appendix A.2] For all $r \in [n]$, $\mathrm{Var}[\varphi_{\geq r}^{\boldsymbol{n}^{\mathrm{red}}}] \leq \frac{16n}{B} \cdot \left( \frac{n}{r^2} + \sum_{t \in [r-1]} \frac{t \cdot \varphi_t^{\boldsymbol{n}}}{r(r-t)} \right)$.*

Plugging Lemma 16 into (5), we have

$$
\begin{aligned}
\mathbb{E}[\|\boldsymbol{\varphi}_{\geq}^{\boldsymbol{n}^{\mathrm{red}}} - \Gamma^B(\boldsymbol{n})\|_1] &\leq O\left(\sqrt{n/B}\right) \cdot \sum_{r \in [n]} \sqrt{\frac{n}{r^2} + \sum_{t \in [r-1]} \frac{t \cdot \varphi_t^{\boldsymbol{n}}}{r(r-t)}} \\
\text{(Cauchy–Schwarz)} \quad &\leq O\left(\sqrt{n/B}\right) \cdot \sqrt{\sum_{r \in [n]} \frac{1}{r}} \sqrt{\sum_{r \in [n]} r \cdot \left( \frac{n}{r^2} + \sum_{t \in [r-1]} \frac{t \cdot \varphi_t^{\boldsymbol{n}}}{r(r-t)} \right)} \\
&= O\left(\sqrt{n \log(n)/B}\right) \sqrt{\sum_{r \in [n]} \left( \frac{n}{r} + \sum_{t \in [r-1]} \frac{t \cdot \varphi_t^{\boldsymbol{n}}}{r-t} \right)} \\
&= O\left(\sqrt{n \log(n)/B}\right) \sqrt{O(n \log n) + \sum_{t \in [n-1]} \sum_{\ell \in [n-t]} \frac{t \cdot \varphi_t^{\boldsymbol{n}}}{\ell}} \\
&\leq O\left(\sqrt{n \log(n)/B}\right) \sqrt{O(n \log n) + O(\log n) \cdot \sum_{t \in [n-1]} t \cdot \varphi_t^{\boldsymbol{n}}} \\
&= O\left(\sqrt{n \log(n)/B}\right) \sqrt{O(n \log n)} \\
&= O(n \log n/\sqrt{B}) \,. \qquad \square
\end{aligned}
$$

**Putting things together.** With all the components ready, we can now prove Theorem 12.

*Proof of Theorem 12.* By Lemma 14, we have

$$
\begin{aligned}
\mathbb{E}[\|\boldsymbol{n} - \hat{\boldsymbol{n}}\|_1] &\leq 4 \cdot \mathbb{E}[\|\Gamma^B(\boldsymbol{n}) - \Gamma^B(\hat{\boldsymbol{n}})\|_1] \\
&\leq 4 \cdot \left( \mathbb{E}[\|\Gamma^B(\boldsymbol{n}) - \boldsymbol{\varphi}_{\geq}^{\hat{\boldsymbol{n}}^{\mathrm{red}}}\|_1 + \|\boldsymbol{\varphi}_{\geq}^{\hat{\boldsymbol{n}}^{\mathrm{red}}} - \Gamma^B(\hat{\boldsymbol{n}})\|_1] \right) \\
&\leq 8 \cdot \left( \mathbb{E}[\|\Gamma^B(\boldsymbol{n}) - \boldsymbol{\varphi}_{\geq}^{\hat{\boldsymbol{n}}^{\mathrm{red}}}\|_1] \right) \qquad \text{(By definition of } \hat{\boldsymbol{n}}\text{)} \\
&\leq 8 \cdot \left( \mathbb{E}[\|\boldsymbol{\varphi}_{\geq}^{\boldsymbol{n}^{\mathrm{red}}} - \boldsymbol{\varphi}_{\geq}^{\hat{\boldsymbol{n}}^{\mathrm{red}}}\|_1 + \|\Gamma^B(\boldsymbol{n}) - \boldsymbol{\varphi}_{\geq}^{\boldsymbol{n}^{\mathrm{red}}}\|_1] \right) \\
&\leq O(\mathbb{E}[\|\boldsymbol{n}^{\mathrm{red}} - \hat{\boldsymbol{n}}^{\mathrm{red}}\|_1] + n \log n/\sqrt{B}) \qquad \text{(From Lemma 15).} \qquad \square
\end{aligned}
$$

# 5 Estimating Symmetric Properties of Discrete Distributions

In this section we show how to use Theorem 12 for the task of estimating symmetric properties of discrete distributions over $[k]$. Here, a distribution property is *symmetric* if it remains unchanged under relabeling of the domain symbols. For example, a notable such property is the Shannon entropy of a distribution $\mathcal{D}$ defined as $H(\mathcal{D}) := \sum_x \mathcal{D}(x) \log \frac{1}{\mathcal{D}(x)}$, a central object in information theory, machine learning, and statistics. If the support is unbounded, estimating $H(\mathcal{D})$ is impossible with any finite number of samples. Our goal is to estimate the entropy of a distribution $\mathcal{D} \in \Delta_k$ up to an additive $\pm\alpha$ error, where $\Delta_k$ denotes the set of all distributions over $[k]$.

One of the key ideas in the literature (e.g., [3]) is to design *low sensitivity* estimators $\hat{f} : \mathcal{X}^n \to \mathbb{R}$ for the desired symmetric distribution property $f : \Delta_k \to \mathbb{R}$. The (non-private) *sample complexity* of a property $f : \Delta_k \to \mathbb{R}$ using estimator $\hat{f}$, denoted by $C_{\hat{f}}(f, \alpha)$, is the smallest number of samples $n$ needed to estimate $f(\mathcal{D})$ upto accuracy $\alpha$ with probability at least 0.9, that is, $\Pr[|\hat{f}(S) - f(\mathcal{D})| > \alpha] < 0.1$.[8] The *sensitivity* of an estimator $\hat{f}$ is $\Delta_{n,\hat{f}}$, which is the smallest value for which it holds for adjacent datasets $S \sim S'$ each with $n$ elements, that $|\hat{f}(S) - \hat{f}(S')| \le \Delta_{n,\hat{f}}$. Let $D_{\hat{f}}(\alpha, \varepsilon) := \min\{n \mid \Delta_{n,\hat{f}} \le 0.1\alpha/E(n,\varepsilon)\}$, for $E(n,\varepsilon)$ defined in Corollary 13.

We will only consider *symmetric* estimators $\hat{f}$, for which we will abuse notation to use $\hat{f}(\boldsymbol{n})$ to denote $\hat{f}(S)$ for any dataset $S$ that corresponds to the anonymized histogram $\boldsymbol{n}$.

**Theorem 17.** *For all $0 < \varepsilon \le O(1), \delta \in (0,1]$, for any symmetric distribution property $f$, and any symmetric estimator $\hat{f}$, there exists an $\varepsilon$-DP mechanism in the pan-private model and $(\varepsilon, \delta)$-DP mechanism in the shuffle DP model, such that $\Pr_{S \sim \mathcal{D}^n}[|\mathcal{M}(S) - f(\mathcal{D})| > \alpha] < 0.2$ with sample complexity*

$$O\left(C_{\hat{f}}\left(f, \frac{\alpha}{2}\right) + D_{\hat{f}}\left(\frac{\alpha}{2}, \varepsilon\right)\right).$$

*Proof.* Let $\boldsymbol{n}$ denote the anonymized histogram corresponding to the sampled dataset $S$. The mechanism $\mathcal{M}$ simply outputs $\hat{f}(\hat{\boldsymbol{n}})$ for $\hat{\boldsymbol{n}}$ returned by Algorithm 2. Clearly the mechanism $\mathcal{M}$ is DP due to the post-processing property.

We have that for a suitable $n = O\left(C_{\hat{f}}\left(f, \frac{\alpha}{2}\right) + D_{\hat{f}}\left(\frac{\alpha}{2}, \varepsilon\right)\right)$, it holds that

$$\Pr\left[\left|\hat{f}(\boldsymbol{n}) - f(\mathcal{D})\right| > \frac{\alpha}{2}\right] \le 0.1 \qquad \text{and} \qquad \Pr\left[\left|\hat{f}(\hat{\boldsymbol{n}}) - \hat{f}(\boldsymbol{n})\right| > \frac{\alpha}{2}\right] \le 0.1,$$

where the first inequality holds by definition of $C_{\hat{f}}(f, \alpha/2)$, and the second inequality holds because $|\hat{f}(\hat{\boldsymbol{n}}) - \hat{f}(\boldsymbol{n})| \le \Delta_{n,\hat{f}} \cdot \|\hat{\boldsymbol{n}} - \boldsymbol{n}\|_1$ and by the guarantee of Theorem 12 and Markov's inequality $\Pr[\|\hat{\boldsymbol{n}} - \boldsymbol{n}\|_1 > 10E(n,\varepsilon)] \le 0.1$. By a union bound, we have $\Pr[|\hat{f}(\hat{\boldsymbol{n}}) - \hat{f}(\boldsymbol{n})| > \alpha] \le 0.44$. $\square$

We get the following sample complexity bounds for private estimation of Shannon entropy in the pan-private and shuffle DP models, as an immediate application of Theorem 17.

**Corollary 18** (Proof in Appendix C)**.** *For all $0 < \varepsilon \le O(1), \delta \in (0,1]$, there exists an $\varepsilon$-DP mechanism in the pan-private model and $(\varepsilon, \delta)$-DP mechanism in the shuffle DP model, that can estimate the entropy of $\mathcal{D} \in \Delta_k$ up to an additive error of $\pm\alpha$ with a sample complexity of*

$$\min_{\lambda \in (0, \frac{1}{2})} \left\{ O\left(\frac{k}{\alpha} + \frac{\log^2 k}{\alpha^2} + \frac{\log^{3.5}(1/(\alpha\varepsilon))}{\alpha^2 \varepsilon^5}\right), \quad O\left(\frac{k}{\lambda^2 \alpha \log k} + \frac{\log^2 k}{\alpha^2} + \left(\frac{\log^{1.5}(1/(\alpha\varepsilon))}{\alpha^2 \varepsilon^5}\right)^{1/(1-2\lambda)}\right) \right\}.$$

These bounds have the same dependence on $k$ as in the work of Acharya et al. [3]. The dependence on $\alpha$ and $\varepsilon$ is slightly worse due to a higher cost of sensitivity in our setting, and the worse dependence on $\varepsilon$ in our guarantees in Corollary 13.

We also derive sample complexity bounds for private estimation of support coverage and support size. The *support coverage* of a distribution $\mathcal{D}$ and an integer $m$ is defined as $S_m(\mathcal{D}) :=$

---

[8]The choice of 0.1 is arbitrary; using the "median trick", we can boost the success probability to $1 - \beta$ with an additional multiplicative $\log(1/\beta)$ more samples.

$\sum_{x \in \text{supp}(\mathcal{D})}(1 - (1 - \mathcal{D}(x))^m)$, i.e., the expected number of distinct elements seen if we draw $m$ i.i.d. samples from $\mathcal{D}$. Here we would like to estimate $S_m(\mathcal{D})$ to within an additive error of $\pm \alpha m$. The *support size* of a discrete distribution $\mathcal{D}$ is the number of atoms $x$ such that $\mathcal{D}(x) > 0$. In general, this is impossible with finite sample since some atom of $\mathcal{D}$ might have an arbritrarily small probability mass. To avoid this, we follow prior work and consider only probability distributions in $\Delta_{\geq 1/K} := \{\mathcal{D} \mid \forall x \in \text{supp}(\mathcal{D}), \mathcal{D}(x) \geq 1/K\}$, i.e., those with non-zero mass of at least $1/K$ at every atom, for some $K$. We defer the details of the exact sample complexity bounds and the proofs to Appendix C.

## 6    Conclusions and Future Directions

In this paper, we give simple algorithms for privately computing anonymized histograms. Our algorithms can be implemented in the shuffle and pan-private models. There are several immediate open questions as discussed below.

Our upper bounds have a dependency of $O(1/\varepsilon^{2.5})$ in the $\ell_1$-error case and $O(1/\varepsilon^{3.5})$ in the $\ell_2^2$-error case; it is unclear if these are tight. Similarly, there is a lower order multiplicative term of $O(\log n)$ and $O(\log^2 n)$ in our $\ell_1$- and $\ell_2^2$-error bounds respectively. Closing these gaps would be an interesting next step; these would also lead to improvements to the sample complexity bounds on private estimation of symmetric distribution properties, such as the Shannon entropy (Corollary 18).

While our analysis of the large domain case relies on the hash function being uniformly random, it is conceivable that this is not required for the approach to work. It will be interesting if this approach, perhaps with some modifications, can also work with a weaker family of hash functions such as a pairwise-independent hash family.

Note also that our shuffle DP algorithm has $\delta > 0$, i.e., approximate-DP. This may not be necessary: there are a couple of recent algorithms for computing histogram with shuffle DP with $\delta = 0$ [28, 19]. Our post-processing approach does not immediately apply to these algorithms because the noise to each count is not a discrete Laplace noise (and in fact is not even an independent additive noise). Adapting our approach to their setting is another interesting research direction.

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
