# Appendix for Anonymized Histograms in Intermediate Privacy Models

## A    Missing Proofs from Section 4

### A.1    Proof of Lemma 14

**Lemma 14.** *[Proof in Appendix A.1]  For $B > 4n$ and all anonymized histograms $\boldsymbol{n}, \boldsymbol{n}'$ with $\|\boldsymbol{n}\|_1, \|\boldsymbol{n}'\|_1 \leq n$,*

$$\|\Gamma^B(\boldsymbol{n}) - \Gamma^B(\boldsymbol{n}')\|_1 \geq \|\boldsymbol{n} - \boldsymbol{n}'\|_1/4 \,.$$

We start with the following observation that $\Gamma^B(\boldsymbol{n})_\ell := \mathbb{E}[\varphi^{\boldsymbol{n}^{\mathrm{red}}}_{\geq \ell}]$ only depends on $\varphi^{\boldsymbol{n}}_{\geq 1}, \ldots, \varphi^{\boldsymbol{n}}_{\geq \ell}$. This is simply because changing an entry of value at least $\ell$ to some other value at least $\ell$ does not change whether, after hashing, the corresponding hash bucket exceeds $\ell$.

**Observation 19.**  $\Gamma^B(\boldsymbol{n})_\ell$ does *not* depend on $\varphi^{\boldsymbol{n}}_{\geq r}$ for any $r > \ell$.

Before can prove Lemma 14, we will also need to show the following lemma, which is even stronger than Lipschitzness.

**Lemma 20.** *Let $\bar{\boldsymbol{n}}, \tilde{\boldsymbol{n}}$ be anonymized histograms such that $\|\tilde{\boldsymbol{n}} - \bar{\boldsymbol{n}}\|_1 = 1$, $\|\bar{\boldsymbol{n}}\|_1 = n$, $\|\tilde{\boldsymbol{n}}\|_1 = n+1$, and $r \in [n]$ be such that $\varphi^{\tilde{\boldsymbol{n}}}_{\geq r} = \varphi^{\bar{\boldsymbol{n}}}_{\geq r} + 1$. Then, we have*

- ▸ $\Gamma^B(\tilde{\boldsymbol{n}})_\ell = \Gamma^B(\bar{\boldsymbol{n}})_\ell$ *for all $\ell < r$.*
- ▸ $\Gamma^B(\tilde{\boldsymbol{n}})_r \geq \Gamma^B(\bar{\boldsymbol{n}})_r + 1 - n/B.$
- ▸ $\sum_{\ell > r} |\Gamma^B(\tilde{\boldsymbol{n}})_\ell - \Gamma^B(\bar{\boldsymbol{n}})_\ell| < n/B.$

*Specifically, these also imply that $\|\Gamma^B(\tilde{\boldsymbol{n}}) - \Gamma^B(\bar{\boldsymbol{n}})\|_1 \leq 1$.*

*Proof.* Let $\bar{\boldsymbol{n}}^{\mathrm{red}}$ and $\tilde{\boldsymbol{n}}^{\mathrm{red}}$ be the random variables as defined at the beginning of Section 4, but with starting anonymized histograms $\bar{\boldsymbol{n}}, \tilde{\boldsymbol{n}}$ respectively. Furthermore, let $j$ be the entry such that $\tilde{n}^{(j)} = \bar{n}^{(j)} + 1$.

Recall that $H$ is our random hash function. Consider $\bar{\boldsymbol{n}}^{\mathrm{red}}(H)$ and $\tilde{\boldsymbol{n}}^{\mathrm{red}}(H)$ resulting from the same hash function $H$. By definition, we have $\Gamma^B(\tilde{\boldsymbol{n}}) - \Gamma^B(\bar{\boldsymbol{n}}) = \mathbb{E}_H[\varphi^{\tilde{\boldsymbol{n}}^{\mathrm{red}}(H)}_{\geq} - \varphi^{\bar{\boldsymbol{n}}^{\mathrm{red}}(H)}_{\geq}]$.

Furthermore, we have $\varphi^{\tilde{\boldsymbol{n}}^{\mathrm{red}}(H)}_{\geq} - \varphi^{\bar{\boldsymbol{n}}^{\mathrm{red}}(H)}_{\geq} = \mathbf{1}_{\tilde{n}^{\mathrm{red}}_{H(j)}}$, where $\tilde{n}^{\mathrm{red}}_{H(j)}$ is the value of the bucket $H(j)$ to which $j$ gets hashed. Note that with probability at least $1 - n/B$, there is no collision with $j$ and therefore $\tilde{n}^{\mathrm{red}}_{H(j)} = r$. Otherwise, with probability at most $n/B$, there is a collision and $\tilde{n}^{\mathrm{red}}_{H(j)} > r$. This implies the desired bounds.    □

We are now ready to prove Lemma 14.

*Proof of Lemma 14.* Let $\boldsymbol{n}^{\min}$ be such that $\varphi^{\boldsymbol{n}^{\min}}_{\geq r} = \min\{\varphi^{\boldsymbol{n}}_{\geq r}, \varphi^{\boldsymbol{n}'}_{\geq r}\}$ for all $r \in \mathbb{N}$. Note that $\|\boldsymbol{n}^{\min} - \boldsymbol{n}\|_1 + \|\boldsymbol{n}^{\min} - \boldsymbol{n}'\|_1 = \|\boldsymbol{n} - \boldsymbol{n}'\|_1$. Assume w.l.o.g. that $\|\boldsymbol{n}^{\min} - \boldsymbol{n}\|_1 \geq \|\boldsymbol{n}^{\min} - \boldsymbol{n}'\|_1$. The previous inequality implies that $\|\boldsymbol{n}^{\min} - \boldsymbol{n}\|_1 \geq \|\boldsymbol{n} - \boldsymbol{n}'\|_1/2$.

Now, let $S := \{r \mid \varphi^{\boldsymbol{n}}_{\geq r} > \varphi^{\boldsymbol{n}^{\min}}_{\geq r}\}$. Consider the following hybrids for $j = 0, \ldots, n$: let $\boldsymbol{n}^j$ be such that $\varphi^{\boldsymbol{n}^j}_{\geq} = (\varphi^{\boldsymbol{n}}_{\geq 1}, \ldots, \varphi^{\boldsymbol{n}}_{\geq j}, \varphi^{\boldsymbol{n}^{\min}}_{\geq j+1}, \ldots, \varphi^{\boldsymbol{n}^{\min}}_{\geq n})$. By Lemma 20 and the definition of $S$, we have

$$\sum_{r \in S} \left( \Gamma^B(\boldsymbol{n}^{j+1})_r - \Gamma^B(\boldsymbol{n}^j)_r \right) \geq \frac{3}{4} \left( \varphi^{\boldsymbol{n}^{j+1}}_{\geq j+1} - \varphi^{\boldsymbol{n}^j}_{\geq j+1} \right),$$

since $\|\boldsymbol{n}^{\min}\|_1 \leq \|\boldsymbol{n}\|_1 \leq n$. By summing the above over all $j = 0, \ldots, n-1$, we have

$$\sum_{r \in S} \left( \Gamma^B(\boldsymbol{n})_r - \Gamma^B(\boldsymbol{n}^{\min})_r \right) \geq \frac{3}{4} \|\boldsymbol{n}^{\min} - \boldsymbol{n}\|_1. \tag{6}$$

Similarly, we can consider the following hybrids for $j = 0, \ldots, n$: let $\boldsymbol{n}'^j$ be such that $\varphi_{\geq}^{\boldsymbol{n}'^j} = (\varphi_{\geq 1}^{\boldsymbol{n}'}, \ldots, \varphi_{\geq j}^{\boldsymbol{n}'}, \varphi_{\geq j+1}^{\boldsymbol{n}^{\min}}, \ldots, \varphi_{\geq n}^{\boldsymbol{n}^{\min}})$. By Lemma 20 and the definition of $S$ (which implies that $\varphi_{\geq r}^{\boldsymbol{n}'} = \varphi_{\geq r}^{\boldsymbol{n}^{\min}}$ for all $r \in S$), we have

$$\sum_{r \in S} \left( \Gamma^B(\boldsymbol{n}'^{j+1})_r - \Gamma^B(\boldsymbol{n}'^j)_r \right) \leq \frac{1}{4} \left( \varphi_{\geq j+1}^{\boldsymbol{n}'^{j+1}} - \varphi_{\geq j+1}^{\boldsymbol{n}'^j} \right).$$

Again, by summing the above over all $j = 0, \ldots, n-1$, we have

$$\sum_{r \in S} \left( \Gamma^B(\boldsymbol{n}')_r - \Gamma^B(\boldsymbol{n}^{\min})_r \right) \leq \frac{1}{4} \| \boldsymbol{n}^{\min} - \boldsymbol{n}' \|_1. \tag{7}$$

Subtracting (7) from (6), we complete the proof as follows:

$$\| \Gamma^B(\boldsymbol{n}) - \Gamma^B(\boldsymbol{n}') \|_1 \geq \sum_{r \in S} \Gamma^B(\boldsymbol{n})_r - \Gamma^B(\boldsymbol{n}')_r \geq \frac{3}{4} \| \boldsymbol{n}^{\min} - \boldsymbol{n} \|_1 - \frac{1}{4} \| \boldsymbol{n}^{\min} - \boldsymbol{n}' \|_1$$

$$\geq \frac{1}{2} \| \boldsymbol{n}^{\min} - \boldsymbol{n} \|_1 \geq \frac{1}{4} \| \boldsymbol{n} - \boldsymbol{n}' \|_1. \qquad \square$$

## A.2 Proof of Lemma 16

**Lemma 16.** *[Proof in Appendix A.2] For all $r \in [n]$, $\mathrm{Var}[\varphi_{\geq r}^{\boldsymbol{n}^{\mathrm{red}}}] \leq \frac{16n}{B} \cdot \left( \frac{n}{r^2} + \sum_{t \in [r-1]} \frac{t \cdot \varphi_t^{\boldsymbol{n}}}{r(r-t)} \right)$.*

*Proof of Lemma 16.* Note that $\varphi_{\geq r}^{\boldsymbol{n}^{\mathrm{red}}}$ is the number of hash buckets whose total value is at least $r$. Let $X$ denote the number of hash buckets whose maximum value hashed to it is at least $r$, i.e., $X := |\{i \in [B] \mid \max_{j \in H^{-1}(i)} n^{(j)} \geq r\}|$. Furthermore, for $t \in [r-1]$, let $Y_t$ denote the number of buckets whose total value is at least $r$ and the maximum value hashed to it is equal to $t$, i.e., $Y_t := |\{i \in [B] \mid \max_{j \in H^{-1}(i)} n^{(j)} = t, \sum_{j \in H^{-1}(i)} n^{(j)} \geq r\}|$.

It is easy to see that $X, Y_1, \ldots, Y_{r-1}$ are negatively correlated. Therefore,

$$\mathrm{Var}[\varphi_{\geq r}^{\boldsymbol{n}^{\mathrm{red}}}] = \mathrm{Var}[X + Y_1 + \cdots + Y_{r-1}] \leq \mathrm{Var}[X] + \sum_{t \in [r-1]} \mathrm{Var}[Y_t].$$

We will next bound each term in the RHS above.

**Bounding $\mathrm{Var}[X]$.** To bound $\mathrm{Var}[X]$, further note that $X = \varphi_{\geq r}^{\boldsymbol{n}} - \sum_{i \in [B]} Z_i$ where $Z_i := \max\{0, |\{j \in H^{-1}(i) \mid n^{(j)} \geq r\}| - 1\}$. It is again easy to see that $Z_i$'s are negatively correlated. Therefore,

$$\mathrm{Var}[X] = \mathrm{Var}\left[ \sum_{i \in [B]} Z_B \right] \leq \sum_{i \in [B]} \mathrm{Var}[Z_i] \leq \sum_{i \in [B]} \mathbb{E}[Z_i^2].$$

It is also easy to verify that, for any $k \in \mathbb{N}$, $\Pr[Z_i = k] = \binom{\varphi_{\geq r}^{\boldsymbol{n}}}{k+1} (1/B)^{k+1} (1 - 1/B)^{\varphi_{\geq r}^{\boldsymbol{n}} - k - 1} \leq (\varphi_{\geq r}^{\boldsymbol{n}}/B)^{k+1}/(k+1)!$. Therefore, we have

$$\mathbb{E}[Z_i^2] \leq \sum_{k \in \mathbb{N}} k^2 \cdot (\varphi_{\geq r}^{\boldsymbol{n}}/B)^{k+1}/(k+1)! \leq 2 \cdot \sum_{k \in \mathbb{N}} (\varphi_{\geq r}^{\boldsymbol{n}}/B)^{k+1}$$

$$\leq 4 \left( \frac{\varphi_{\geq r}^{\boldsymbol{n}}}{B} \right)^2 \leq 4 \left( \frac{n}{Br} \right)^2 = \frac{4}{r^2} \cdot \left( \frac{n}{B} \right)^2.$$

Plugging this back into the previous inequality, we have

$$\mathrm{Var}[X] \leq \frac{4}{r^2} \cdot \frac{n^2}{B}.$$

**Bounding $\text{Var}[Y_t]$.** We may write $Y_t$ as $\sum_{i \in [B]} W_i$ where $W_i := \mathbf{1}[\max_{j \in H^{-1}(i)} n^{(j)} = t, \sum_{j \in H^{-1}(i)} n^{(j)} \geq r]$. Again, the $W_i$'s are negatively correlated. Therefore, we have

$$\text{Var}[Y_t] = \text{Var}\left[\sum_{i \in [B]} W_i\right] \leq \sum_{i \in [B]} \text{Var}[W_i] \leq \sum_{i \in [B]} \Pr[W_i = 1], \tag{8}$$

where the last inequality uses the fact that $W_i$ is a Bernoulli r.v.

For each $t$, let $S_t \subseteq \mathbb{N}$ denote the set of indices $j$ for which $n^{(j)} = t$. Furthermore, let $S_{\leq t} := S_1 \cup \cdots \cup S_t$. Notice that

$$\Pr[W_i = 1] = \Pr\left[\max_{j \in H^{-1}(i)} n^{(j)} = t \text{ and } \sum_{j \in H^{-1}(i)} n^{(j)} \geq r\right]$$

$$= \Pr\left[\max_{j \in H^{-1}(i)} n^{(j)} = t \text{ and } \sum_{j \in H^{-1}(i) \cap S_{\leq t}} n^{(j)} \geq r\right]$$

$$= \Pr\left[\exists j^* \in S_t \text{ s.t } H(j^*) = i \text{ and } \sum_{j \in H^{-1}(i) \cap S_{\leq t}} n^{(j)} \geq r\right]$$

$$(\text{Union bound}) \leq \sum_{j^* \in S_t} \Pr\left[H(j^*) = i \text{ and } \sum_{j \in H^{-1}(i) \cap S_{\leq t}} n^{(j)} \geq r\right]$$

$$= \sum_{j^* \in S_t} \Pr\left[H(j^*) = i \text{ and } \sum_{j \in H^{-1}(i) \cap (S_{\leq t} \setminus \{j^*\})} n^{(j)} \geq r - t\right]$$

$$(\text{Independence of hash values}) = \sum_{j^* \in S_t} \Pr[H(j^*) = i] \Pr\left[\sum_{j \in H^{-1}(i) \cap (S_{\leq t} \setminus \{j^*\})} n^{(j)} \geq r - t\right]$$

$$= \sum_{j^* \in S_t} \frac{1}{B} \cdot \Pr\left[\sum_{j \in H^{-1}(i) \cap (S_{\leq t} \setminus \{j^*\})} n^{(j)} \geq r - t\right]$$

$$\leq \sum_{j^* \in S_t} \frac{1}{B} \cdot \Pr\left[\sum_{j \in H^{-1}(i) \cap S_{\leq t}} n^{(j)} \geq r - t\right]$$

$$= \frac{\varphi_t^{\boldsymbol{n}}}{B} \cdot \Pr\left[\sum_{j \in H^{-1}(i) \cap S_{\leq t}} n^{(j)} \geq r - t\right]$$

$$= \frac{\varphi_t^{\boldsymbol{n}}}{B} \cdot \Pr\left[\sum_{j \in S_{\leq t}} n^{(j)} \cdot U_j \geq r - t\right], \tag{9}$$

where $U_j$ denote the indicator $\mathbf{1}[H(j) = i]$.

We will next bound the last term based on two cases:

▶ Case I: $t \geq r/2$. Note that

$$\mathbb{E}\left[\sum_{j \in S_{\leq t}} n^{(j)} \cdot U_j\right] = \left(\sum_{j \in S_{\leq t}} n^{(j)}\right) \cdot \frac{1}{B} \leq \frac{n}{B}.$$

Therefore, we may apply Markov's inequality to get

$$\Pr\left[\sum_{j \in S_{\leq t}} n^{(j)} \cdot U_j \geq r - t\right] \leq \frac{n}{B(r - t)} \leq \frac{2n}{B} \cdot \frac{t}{r(r - t)},$$

where the latter inequality follows from $t \geq r/2$.

▶ Case II: $t < r/2$. Since $U_j$'s are independent, we can also compute the variance of the sum as

$$\mathrm{Var}\left[\sum_{j\in S_{\leq t}} n^{(j)} \cdot U_j\right] = \sum_{j\in S_{\leq t}} \mathrm{Var}[n^{(j)} \cdot U_j] \leq \sum_{j\in S_{\leq t}} \frac{(n^{(j)})^2}{B}$$

$$= \frac{1}{B}\sum_{\ell\in[t]} \ell^2 \cdot \varphi_\ell^{\boldsymbol{n}} \leq \frac{t}{B}\sum_{\ell\in[t]} \ell \cdot \varphi_\ell^{\boldsymbol{n}} \leq \frac{tn}{B}.$$

Note also that we have $r - t - n/B > (r-t)/2 \geq r/4$, where we used $t < r/2$. We may now apply Chebyshev's inequality to get

$$\Pr\left[\sum_{j\in S_{\leq t}} n^{(j)} \cdot U_j \geq r - t\right] \leq \frac{tn/B}{(r/4)^2} \leq \frac{16n}{B} \cdot \frac{t}{r(r-t)}.$$

Therefore, in both cases, we have $\Pr\left[\sum_{j\in S_{\leq t}} n^{(j)} \cdot U_j \geq r - t\right] \leq \frac{16n}{B} \cdot \frac{t}{n-t}$. Combining this with (8) and (9), we get

$$\mathrm{Var}[Y_t] \leq B \cdot \frac{\varphi_t^{\boldsymbol{n}}}{B} \cdot \frac{16n}{B} \cdot \frac{t}{r(r-t)} = \frac{16n}{B} \cdot \frac{t \cdot \varphi_t^{\boldsymbol{n}}}{r(r-t)}.$$

By summing up our bounds on $X$ and $Y_t$'s, we arrive at the desired bound

$$\mathrm{Var}[\varphi_{\geq r}^{\boldsymbol{n}^{\mathrm{red}}}] \leq \frac{16n}{B} \cdot \left(\frac{n}{r^2} + \sum_{t\in[r-1]} \frac{t \cdot \varphi_t^{\boldsymbol{n}}}{r(r-t)}\right). \qquad \square$$

# B  Extension to $\ell_2^2$-error

A different error measure used in [34] is the $\ell_2^2$-error, defined as $\|\boldsymbol{n} - \hat{\boldsymbol{n}}\|_2^2 = \sum_{j\in[D]}(n^{(j)} - \hat{n}^{(j)})^2$. In this section, we show the flexibility of our method by showing that we can get an error of $\widetilde{O}_\varepsilon(\sqrt{n})$, nearly matching that of [34].

## B.1  Small Domain Case

We start with the setting $D \leq \widetilde{O}(n)$ which does not require hashing.

The idea is to constrain our estimate anonymized histogram such that the $\ell_\infty$-distance from the discrete Laplace-noised histogram is not too large. A full description is presented in Algorithm 3.

---

**Algorithm 3** Anonymized Histogram Estimator w.r.t. $\ell_2^2$ loss.

---

**Input:** Discrete Laplace-noised histogram $\boldsymbol{h}'$, i.e., $h'_j \sim h_j + \mathrm{DLap}(p)$
**Parameter:** $\gamma = 10\log(2nD)/\log(1/p)$

**for** $r \in [n]$ **do**
 $\hat{\varphi}_{\geq r} \leftarrow \sum_{j\in[D]} f(h'_j - r)$, where $f$ is defined in (1)
**return** $\hat{\boldsymbol{n}}$ that minimizes $\|\boldsymbol{\varphi}_{\geq}^{\hat{\boldsymbol{n}}} - \hat{\boldsymbol{\varphi}}_{\geq}\|_1$ subject to $\|\boldsymbol{n}_{\boldsymbol{h}'} - \hat{\boldsymbol{n}}\|_\infty \leq \gamma$ and $\|\hat{\boldsymbol{n}}\|_1 = n$. (If no such $\hat{\boldsymbol{n}}$ exists, output the all-zeros histogram.)

---

**Theorem 21.** *Let $\hat{\boldsymbol{n}}$ be the output of Algorithm 3. We have*

$$\mathbb{E}[\|\hat{\boldsymbol{n}} - \boldsymbol{n}\|_2^2] \leq O\left(\frac{\log(nD)}{\log(1/p)} \cdot \sqrt{C_p(n+D)\log n}\right) + O(1).$$

Plugging in $p = e^{-\varepsilon/2}$ (sufficient for $\varepsilon$-DP), we have the bound of $O(\sqrt{(n+D)\log n} \cdot \log(nD)/\varepsilon^{3.5})$ for $\varepsilon \leq 1$.

*Proof.* First, the standard concentration of the noise implies that

$$\Pr[\|\boldsymbol{n} - \boldsymbol{n_{h'}}\|_\infty > \gamma] \leq 0.1/n^2.$$

We thus have

$$\mathbb{E}[\|\hat{\boldsymbol{n}} - \boldsymbol{n}\|_2^2] \leq \Pr[\|\boldsymbol{n} - \boldsymbol{n_{h'}}\|_\infty \leq \gamma] \cdot \mathbb{E}[\|\hat{\boldsymbol{n}} - \boldsymbol{n}\|_2^2 \mid \|\boldsymbol{n} - \boldsymbol{n_{h'}}\|_\infty \leq \gamma] + \Pr[\|\boldsymbol{n} - \boldsymbol{n_{h'}}\|_\infty > \gamma] \cdot n^2$$

$$\leq \Pr[\|\boldsymbol{n} - \boldsymbol{n_{h'}}\|_\infty \leq \gamma] \cdot \mathbb{E}[\|\hat{\boldsymbol{n}} - \boldsymbol{n}\|_2^2 \mid \|\boldsymbol{n} - \boldsymbol{n_{h'}}\|_\infty \leq \gamma] + O(1).$$

Now, recall that $\|\boldsymbol{n} - \hat{\boldsymbol{n}}\|_2^2 \leq \|\boldsymbol{n} - \hat{\boldsymbol{n}}\|_\infty \cdot \|\boldsymbol{n} - \hat{\boldsymbol{n}}\|_1$. Plugging this into the above, we get

$$\mathbb{E}[\|\hat{\boldsymbol{n}} - \boldsymbol{n}\|_2^2] \leq \Pr[\|\boldsymbol{n} - \boldsymbol{n_{h'}}\|_\infty \leq \gamma] \cdot \mathbb{E}[\|\hat{\boldsymbol{n}} - \boldsymbol{n}\|_1 \cdot \|\hat{\boldsymbol{n}} - \boldsymbol{n}\|_\infty \mid \|\boldsymbol{n} - \boldsymbol{n_{h'}}\|_\infty \leq \gamma] + O(1)$$

$$\leq \Pr[\|\boldsymbol{n} - \boldsymbol{n_{h'}}\|_\infty \leq \gamma] \cdot \mathbb{E}[\|\hat{\boldsymbol{n}} - \boldsymbol{n}\|_1 \cdot 2\gamma \mid \|\boldsymbol{n} - \boldsymbol{n_{h'}}\|_\infty \leq \gamma] + O(1)$$

$$\leq 2\gamma \cdot \Pr[\|\boldsymbol{n} - \boldsymbol{n_{h'}}\|_\infty \leq \gamma] \cdot \mathbb{E}[2\|\boldsymbol{\varphi}_\geq^{\boldsymbol{n}} - \hat{\boldsymbol{\varphi}}_\geq\|_1 \mid \|\boldsymbol{n} - \boldsymbol{n_{h'}}\|_\infty \leq \gamma] + O(1)$$

$$\leq 4\gamma \cdot \mathbb{E}[\|\boldsymbol{\varphi}_\geq^{\boldsymbol{n}} - \hat{\boldsymbol{\varphi}}_\geq\|_1] + O(1)$$

$$\leq 4\gamma \cdot O\left(\sqrt{C_p(n+D)\log n}\right) + O(1),$$

$$= O\left(\frac{\log(nD)}{\log(1/p)} \cdot \sqrt{C_p(n+D)\log n}\right) + O(1),$$

where the second inequality follows from the constraint on $\hat{\boldsymbol{n}}$ and the last inequality follows from our analysis in the $\ell_1$-error case (Theorem 9). $\square$

## B.2 Large Domain Case

We next move on to the case $D \gg n$, which will require random hashing.

### B.2.1 Barrier to Extending to the $\ell_2^2$-error

It turns out that, unlike the $\ell_1$-error case, using a single noisy hashed histogram with $B = \widetilde{O}(n)$ is not sufficient for us to get $\widetilde{O}(\sqrt{n})$ $\ell_2^2$-error. Before we provide the fix for this, let us briefly sketch why this is the case.

Let $c = \lceil 10\sqrt{B} \rceil$ and $q = \lfloor n/c \rfloor$. Consider the following two datasets (before hashing):

▶ There are $c$ items, each with value $q$.
▶ There are $c - 2$ items each with value $q$ and one additional item with value $2q$.

It is not hard to see (from birthday paradox) that, after randomly hashing into $B$ buckets, it is impossible to distinguish the two cases with advantage more than 0.1 (even without any discrete Laplace noise). This implies that the expected $\ell_2^2$-error must be at least $\Omega(q^2) = \Omega(n^2/B)$. For $B = \widetilde{O}(n)$, this is at least $\widetilde{\Omega}(n)$.

### B.2.2 Two-Hash Approach

We now sketch an approach for large $D$. Here we would use two hashes. The first is with $B_1 = \widetilde{O}(n)$ buckets similar to before and but the second with a much larger, say, $B_2 = O(n^4)$ buckets. We then use the $\ell_1$ approach on the first hash with the additional $\ell_\infty$-constraint on the second hash. The main point here is that w.h.p. there would be no collision at all in the second hash; therefore, the $\ell_\infty$-constraint will be valid. A similar analysis to the small domain case shows that this only adds $O_\varepsilon(\log n)$ multiplicative overhead on the expected $\ell_2^2$-error (compared to the $\ell_1$-error).

More precisely, our approach is presented in Algorithm 4.

**Theorem 22.** *Let $\hat{n}$ be the output of Algorithm 4. We have*

$$\mathbb{E}[\|\hat{\boldsymbol{n}} - \boldsymbol{n}\|_2^2] \leq O\left(\frac{\log(nB_2)}{\log(1/p)} \cdot \left(\sqrt{C_p(n+B_1)\log n} + \frac{n\log n}{\sqrt{B_1}}\right)\right) + O\left(\frac{n^4}{B_2} + 1\right).$$

Plugging in $p = e^{-\varepsilon/4}$ (sufficient for $\varepsilon$-DP), $B_1 = n\sqrt{\log n}$ and $B_2 = n^4$, we have the bound of $O(\sqrt{n}(\log n)^{7/4}/\varepsilon^{3.5})$ for $\varepsilon \leq 1$.

---

**Algorithm 4** Anonymized Histogram Estimator w.r.t. $\ell_2^2$ loss, for large domains.

---

**Input:** Two discrete Laplace-noised histograms $\tilde{\boldsymbol{h}}^{1,\mathrm{red}}$ and $\tilde{\boldsymbol{h}}^{2,\mathrm{red}}$ given as
- ▶ $\tilde{h}_j^{1,\mathrm{red}} \sim h_j^{1,\mathrm{red}} + \mathrm{DLap}(p)$ where the random hash has $B_1$ buckets, and
- ▶ $\tilde{h}_j^{2,\mathrm{red}} \sim h_j^{2,\mathrm{red}} + \mathrm{DLap}(p)$ where the random hash has $B_2$ buckets.

**Parameter:** $\gamma = 10\log(2nB_2)/\log(1/p)$

Compute an estimate $\hat{\boldsymbol{n}}^1$ by running Algorithm 2 on $\tilde{\boldsymbol{h}}^1$.
**return** $\hat{\boldsymbol{n}}$ that minimizes $\|\hat{\boldsymbol{n}} - \hat{\boldsymbol{n}}^1\|_1$ subject to $\|\boldsymbol{n}_{\tilde{\boldsymbol{h}}^{2,\mathrm{red}}} - \hat{\boldsymbol{n}}\|_\infty \leq \gamma$ and $\|\hat{\boldsymbol{n}}\|_1 = n$. (If no such $\hat{\boldsymbol{n}}$ exists, just output the all-zero histogram.)

---

*Proof.* The probability that there is any collision in the second hash is at most $n^2/B_2$. Furthermore, standard concentration inequality of the noise implies that the noise added to any of $h_j^{2,\mathrm{red}}$ is greater than $\gamma$ is at most $0.1/n^2$. Therefore, by a union bound, we have

$$\Pr[\|\boldsymbol{n} - \boldsymbol{n}_{\tilde{\boldsymbol{h}}^{2,\mathrm{red}}}\|_\infty > \gamma] \leq n^2/B_2 + 0.1/n^2.$$

The remaining analysis is now similar to that of Theorem 21. Specifically, we have

$$\begin{aligned}
&\mathbb{E}[\|\hat{\boldsymbol{n}} - \boldsymbol{n}\|_2^2] \\
&\leq \Pr[\|\boldsymbol{n} - \boldsymbol{n}_{\tilde{\boldsymbol{h}}^{2,\mathrm{red}}}\|_\infty \leq \gamma] \cdot \mathbb{E}[\|\hat{\boldsymbol{n}} - \boldsymbol{n}\|_1 \cdot \|\hat{\boldsymbol{n}} - \boldsymbol{n}\|_\infty \mid \|\boldsymbol{n} - \boldsymbol{n}_{\tilde{\boldsymbol{h}}^{2,\mathrm{red}}}\|_\infty \leq \gamma] \\
&\qquad + \Pr[\|\boldsymbol{n} - \boldsymbol{n}_{\tilde{\boldsymbol{h}}^{2,\mathrm{red}}}\|_\infty > \gamma] \cdot n^2 \\
&\leq \Pr[\|\boldsymbol{n} - \boldsymbol{n}_{\tilde{\boldsymbol{h}}^{2,\mathrm{red}}}\|_\infty \leq \gamma] \cdot \mathbb{E}[2\|\hat{\boldsymbol{n}}^1 - \boldsymbol{n}\|_1 \cdot (2\gamma) \mid \|\boldsymbol{n} - \boldsymbol{n}_{\tilde{\boldsymbol{h}}^{2,\mathrm{red}}}\|_\infty \leq \gamma] + n^4/B_2 + 0.1 \\
&\leq (4\gamma) \cdot \mathbb{E}[\|\hat{\boldsymbol{n}}^1 - \boldsymbol{n}\|_1] + n^4/B_2 + 0.1 \\
&\leq O\left(\gamma \cdot \left(\sqrt{C_p(n+B_1)\log n} + \sqrt{n^2/B_1}\log n\right) + n^4/B_2 + 1\right),
\end{aligned}$$

where the last inequality follows from Theorem 12. $\qquad\qquad\square$

## C Missing Proof from Section 5

In this section, we prove sample complexity bounds for estimating symmetric distribution properties. In addition to entropy (Corollary 18), we also show sample complexity bounds for support coverage (Corollary 23) and support size (Corollary 24).

### C.1 Entropy

*Proof of Theorem 17.* Acharya et al. [3] consider two estimators for entropy estimation.

- ▶ The first estimator $\hat{H}$ they study is the entropy of the empirical distribution. This has a non-private sample complexity of $C_{\hat{H}}(H, \alpha) = O\left(\frac{k}{\alpha} + \frac{\log^2 k}{\alpha^2}\right)$. The sensitivity of this estimator is $O(\log n/n)$. Thus, we have $D_{\hat{H}}(\alpha, \varepsilon)$ is the smallest $n$ that satisfies

$$\frac{\log n}{n} \cdot \frac{\sqrt{n}(\log n)^{3/4}}{\varepsilon^{2.5}} \lesssim \alpha.$$

  And so, $D_{\hat{H}}(\alpha, \varepsilon) \leq O(\frac{\log^{3.5}(1/(\alpha\varepsilon))}{\alpha^2\varepsilon^5})$. Thus, plugging this into Theorem 17 yields a sample complexity of $O\left(\frac{k}{\alpha} + \frac{\log^2 k}{\alpha^2} + \frac{\log^{3.5}(1/(\alpha\varepsilon))}{\alpha^2\varepsilon^5}\right)$.

- ▶ The second estimator $\hat{H}$ they study is based on the prior work of [2], which for any $\lambda \in (0,1)$ has a non-private sample complexity of $C_{\hat{H}}(H, \alpha) = O\left(\frac{k}{\lambda^2\alpha\log k} + \frac{\log^2 k}{\alpha^2}\right)$. The sensitivity of this estimator is $1/n^{1-\lambda}$. Thus, we have $D_{\hat{H}}(\alpha, \varepsilon)$ is the smallest $n$ that satisfies

$$\frac{1}{n^{1-\lambda}} \cdot \frac{\sqrt{n}(\log n)^{3/4}}{\varepsilon^{2.5}} \lesssim \alpha.$$

And so, $D_{\hat{H}}(\alpha, \varepsilon) \leq O\left(\left(\frac{\log^{1.5}(1/(\alpha\varepsilon))}{\alpha^2\varepsilon^5}\right)^{1/(1-2\lambda)}\right)$. Thus, plugging this into Theorem 17 yields a sample complexity of $O\left(\frac{k}{\lambda^2\alpha\log k} + \frac{\log^2 k}{\alpha^2} + \left(\frac{\log^{1.5}(1/(\alpha\varepsilon))}{\alpha^2\varepsilon^5}\right)^{1/(1-2\lambda)}\right)$. $\qquad\square$

## C.2  Support Coverage

The *support coverage* of a distribution $\mathcal{D}$ and an integer $m$ is defined as $S_m(\mathcal{D}) := \sum_{x\in\text{supp}(\mathcal{D})}(1 - (1 - \mathcal{D}(x))^m)$, i.e., the expected number of distinct elements seen if we draw $m$ i.i.d. samples from $\mathcal{D}$. Here we would like to estimate $S_m(\mathcal{D})$ to within an additive error of $\pm\alpha m$.

Our bounds are stated below. Note that, in the case of large $m$, our sample complexity is asymptotically the same as that of [3] but our dependency on $\alpha, \varepsilon$ is worse in the case of smaller $m$.

**Corollary 23** (Support Coverage). *For all $0 < \varepsilon \leq O(1), \delta \in (0,1]$, there exists an $(\varepsilon, \delta)$-DP mechanism, in the pan-private and shuffle DP models, that can estimate $S_m(\mathcal{D})$ up to an additive error of $\pm\alpha m$ with a sample complexity of*

$$\begin{cases} O\left(\frac{m\log(1/\alpha)}{\log(\alpha m\varepsilon)}\right) & \text{if } m \geq \Omega\left(\frac{\log^{1.5}(1/(\alpha\varepsilon))}{\alpha^2\varepsilon^5}\right), \\ O\left(\frac{\log^{1.5}(1/(\alpha\varepsilon))}{\alpha^2\varepsilon^5}\right) & \text{if } m \leq O\left(\frac{\log^{1.5}(1/(\alpha\varepsilon))}{\alpha^2\varepsilon^5}\right). \end{cases}$$

*Proof.* Throughout the proof, it will be simpler to think of estimating $\frac{S_m(\mathcal{D})}{m}$ to within an additive error of $\pm\alpha$. Acharya et al. [3] consider two estimators for support coverage estimation.

▶ **Sparse Case.** Assuming that $m \geq \Omega\left(\frac{\log^2(1/(\alpha\varepsilon))}{\alpha^2\varepsilon^5}\right)$. When $m \geq 2n$, Acharya et al. [3] (based on an earlier work [39]) gave an estimator $\hat{S}_m$, parameterized by $r = \log(3/\alpha)$ and $t = m/n - 1$, that has a non-private sample complexity of $C_{\hat{S}_n}(S_m, \alpha) = O\left(\frac{m\log(1/\alpha)}{\log m}\right)$. The sensitivity of this estimator is $\frac{1+e^{r(t-1)}}{m} = \frac{1+e^{\log(3/\alpha)(m/n-2)}}{m}$. Thus, we have $D_{\hat{S}_m}(\alpha, \varepsilon)$ is the smallest $n$ that satisfies

$$\frac{1 + e^{\log(3/\alpha)(m/n-2)}}{m} \cdot \frac{\sqrt{n}(\log n)^{3/4}}{\varepsilon^{2.5}} \leq \alpha$$

And so, $D_{\hat{S}_m}(\alpha, \varepsilon) \leq O\left(\frac{m\log(1/\alpha)}{\log(\alpha m\varepsilon)}\right)$. Thus, plugging this into Theorem 17 gives rise to a sample complexity of $O\left(\frac{m\log(1/\alpha)}{\log(\alpha m\varepsilon)}\right)$.

▶ **Dense Case.** The second estimator $\hat{S}_m$ they study works when $n$ is a multiple of $m$. We remark that, although the estimator as described in [3] does not seem to fit our setting, it can be viewed in our framework as follows. The examples are divided in to $n/m$ batches each of size $m$. We then build a histogram $\boldsymbol{h}$ on $\text{supp}(\mathcal{D}) \times [n/m]$ where each example in batch $i$ appends $i$ to its original item. The final estimate is then $\varphi^{\boldsymbol{h}}_{\geq 1}/n$. It was shown in [3] that the non-private sample complexity is $C_{\hat{S}_m}(S_m, \alpha) = O\left(1/\alpha^2\right)$.

The sensitivity of this estimator is $1/n$. Thus, we have $D_{\hat{S}_m}(\alpha, \varepsilon)$ is the smallest $n$ that satisfies

$$\frac{1}{n} \cdot \frac{\sqrt{n}(\log n)^{3/4}}{\varepsilon^{2.5}} \leq \alpha$$

And so, $D_{\hat{S}_m}(\alpha, \varepsilon) \leq O\left(\frac{\log^{1.5}(1/(\alpha\varepsilon))}{\alpha^2\varepsilon^5}\right)$. Thus, plugging this into Theorem 17 gives rise to a sample complexity of $O\left(\max\left\{m, \frac{\log^{1.5}(1/(\alpha\varepsilon))}{\alpha^2\varepsilon^5}\right\}\right)$, where the $m$ part comes from our constraint that $n \geq m$. $\qquad\square$

## C.3  Support Size

Finally, we consider the problem of estimating the support size of $\mathcal{D}$. In general, this is impossible with finite sample since some atom of $\mathcal{D}$ might have an arbitrarily small probability mass. To avoid this, we follow prior work and consider only probability distributions in $\Delta_{\geq 1/K} := \{\mathcal{D} \mid \forall x \in \text{supp}(\mathcal{D}), \mathcal{D}(x) \geq 1/K\}$, i.e., those with non-zero mass of at least $1/K$ at every atom, for some $K$.

Orlitsky et al. [39] proved that the support coverage with $m \geq \Omega(K \cdot \log(3/\alpha))$ is a good estimate to the support size of any $\mathcal{D} \in \Delta_{\geq 1/K}$ to within an error of $\pm \alpha K$. In particular, they showed that if $m \geq K \log(3/\alpha)$, then for any $\mathcal{D} \in \Delta_{1/K}$, it holds that $|S_m(\mathcal{D}) - S(\mathcal{D})| \leq \alpha K/3$. Combining this with Corollary 24, we immediately get the following bounds for the latter.

**Corollary 24** (Support Size). *For all $0 < \varepsilon \leq O(1), \delta \in (0,1]$, there exists an $(\varepsilon, \delta)$-DP mechanism, in the pan-private and shuffle DP settings, that can estimate the support size of $\mathcal{D} \in \Delta_{\geq 1/K}$ up to an additive error of $\pm \alpha K$ with a sample complexity of*

$$\begin{cases} O\left(\frac{K \log^2(1/\alpha)}{\log(\alpha K \varepsilon)}\right) & \text{if } K \geq \Omega\left(\frac{\log^{1.5}(1/(\alpha\varepsilon))}{\alpha^2 \varepsilon^5 \log(1/\alpha)}\right), \\ O\left(\frac{\log^{1.5}(1/(\alpha\varepsilon))}{\alpha^2 \varepsilon^5}\right) & \text{if } K \leq O\left(\frac{\log^{1.5}(1/(\alpha\varepsilon))}{\alpha^2 \varepsilon^5 \log(1/\alpha)}\right). \end{cases}$$

# D  Fast Post-Processing Algorithms

So far we have not focused on the running time of our post-processing algorithms. Nonetheless, it is not hard to see that our algorithms run in polynomial time. Below, we show that our algorithms can even be made to run in $\widetilde{O}_\varepsilon(D + n)$-time for the $\ell_1$-error case.

## D.1  Small Domain Case

We start with how to implement Algorithm 1 in $O(D + n \log n)$ time. The first observation is that the expression for each $\hat{\varphi}_{\geq r}$ can be easily computed in constant time if we precompute the number of buckets above a certain threshold beforehand. The second observation is that the last step is exactly the same as the so-called $\ell_1$-*isotonic regression* problem (by viewing the cumulative prevalence $\varphi_{\geq 1}, \ldots, \varphi_{\geq n}$ as the variables), for which an $O(n \log n)$-algorithm is known [42]. A full description is given in Algorithm 5.

---

**Algorithm 5** Efficient Anonymized Histogram Estimator w.r.t. $\ell_1$ loss.

---

**Input:** Discrete Laplace-noised histogram $\boldsymbol{h}'$, i.e., $h'_j \sim h_j + \text{DLap}(p)$

$c_0, \ldots, c_n \leftarrow 0$                                                             {Counts for hash values}
**for** $j \in [D]$ **do**
    $v \leftarrow \max\{\min\{h'_j, n\}, 0\}$                           {Clip so that the value is between $0, n$}
    $c_v \leftarrow c_v + 1$
$c_{\geq 0}, \ldots, c_{\geq n+1} \leftarrow 0$                               {Counts for hash values above a certain threshold.}
**for** $r = n, \ldots, 0$ **do**
    $c_{\geq r} \leftarrow c_{\geq r+1} + c_r$
**for** $r \in [n]$ **do**
    $\hat{\varphi}_{\geq r} \leftarrow c_{\geq r+1} + \left(1 + \frac{p}{(1-p)^2}\right) c_r + \left(-\frac{p}{(1-p)^2}\right) c_{r-1}$
Find $\hat{\boldsymbol{n}}$ that minimizes $\|\boldsymbol{\varphi}_{\geq}^{\hat{\boldsymbol{n}}} - \hat{\boldsymbol{\varphi}}_{\geq}\|_1$ using $\ell_1$-isotonic regression algorithm
**return** $\hat{\boldsymbol{n}}$

---

## D.2  Large Domain Case

The large domain case is more complicated, as solving the optimization problem $\|\Gamma^B(\hat{\boldsymbol{n}}) - \boldsymbol{\varphi}_{\geq}^{\hat{\boldsymbol{n}}^{\mathrm{red}}}\|_1$ does not seem to be as simple as isotonic regression. Nonetheless, we give a modified algorithm below that takes $\widetilde{O}_\varepsilon(D + n \log n)$ time and has a similar error bound (within a polylogarithmic factor). The idea is to use two noisy histograms in a similar manner as we did for $\ell_2^2$-error (Algorithm 4): One noisy histogram (without hashing) is used to determine the high-value counts, whereas the other noisy histogram, with hashing, is used to determine the low-value counts.

### D.2.1  Computing $\Gamma^B$

Recall from Observation 19 that $\Gamma^B(\boldsymbol{n})_\ell$ does not depend on $\varphi_{\geq r}^{\boldsymbol{n}}$ for any $r > \ell$. Therefore, we may view $\Gamma^B(\boldsymbol{n})_\ell$ as a function of $\varphi_{\geq 1}^{\boldsymbol{n}}, \ldots, \varphi_{\geq \ell}^{\boldsymbol{n}}$. We overload the notion and write $\Gamma^B(\varphi_{\geq 1}^{\boldsymbol{n}}, \ldots, \varphi_{\geq \ell}^{\boldsymbol{n}})_\ell$ to denote $\Gamma^B(\boldsymbol{n})_\ell$ of a corresponding histogram $\boldsymbol{n}$. We start by giving an algorithm for computing

such a value. Recall that $\Gamma^B(\boldsymbol{n})_\ell$ is defined as the expected number of hash buckets whose total value is at least $\ell$. Let $\Xi_{\geq i}(\boldsymbol{n})$ denote the probability that a fixed hash bucket has total value at least $i$. By linearity of expectation, we have $\Gamma^B(\boldsymbol{n})_\ell = B \cdot \Xi_{\geq \ell}(\boldsymbol{n})$. Below, we keep updating $\Xi_{\geq i}$ as we include one more histogram value, using the fact that the hash function puts this histogram value in a random hash bucket.

---

**Algorithm 6** Computation of $\Gamma^B(\varphi_{\geq 1}, \ldots, \varphi_{\geq \ell})_\ell$.

---

**Input:** $\varphi_{\geq 1}, \ldots, \varphi_{\geq \ell}$, cumulative prevalence of a histogram

$\varphi_{\geq \ell+1} \leftarrow 0.$             {For notational convenience.}
$\Xi_{\geq 0} \leftarrow 1$ and $\Xi_{\geq 1}, \ldots, \Xi_{\geq \ell} \leftarrow 0$             {Initialize $\Xi$ values}
**for** $r = 1, \ldots, \ell$ **do**
   **for** $j = 1, \ldots, \varphi_{\geq r} - \varphi_{\geq r+1}$ **do**             {Add a new histogram entry of value $r$}
      **for** $k = \ell, \ldots, 0$ **do**
         $\Xi_{\geq k} \leftarrow \frac{B-1}{B} \cdot \Xi_{\geq k} + \frac{1}{B} \cdot \Xi_{\geq \max\{k-r,0\}}$
**return** $B \cdot \Xi_{\geq \ell}$

---

Since $\varphi_{\geq 1}^{\boldsymbol{n}} \leq n$, there are at most $n$ pairs $(r, j)$ considered by the algorithm. Therefore, the total runtime of Algorithm 6 is $O(n\ell)$.

### D.2.2 Fast Post-processing Algorithm

Our fast post-processing algorithm for large domains is stated below (Algorithm 7). For convenience, we use the convention that $\varphi_{\geq 0}^{\boldsymbol{n}} = n$.

---

**Algorithm 7** Efficient Anonymized Histogram Estimator w.r.t. $\ell_1$ loss, for large domains.

---

**Input:** Two discrete Laplace-noised histograms $\tilde{\boldsymbol{h}}^1$ and $\tilde{\boldsymbol{h}}^{2,\mathrm{red}}$ given as
    ▶ $\tilde{h}_j^1 \sim h_j + \mathrm{DLap}(p)$ (without any hashing), and
    ▶ $\tilde{h}_j^{2,\mathrm{red}} \sim h_j^{2,\mathrm{red}} + \mathrm{DLap}(p)$ where the random hash has $B$ buckets.
**Parameter:** $m = \lceil 10 \log D / \log(1/p) \rceil$

Compute an estimate $\hat{\boldsymbol{n}}^1$ of $\boldsymbol{n}$ using Algorithm 1
Compute an estimate $\hat{\boldsymbol{n}}^{2,\mathrm{red}}$ of $\boldsymbol{n}^{2,\mathrm{red}}$ using Algorithm 1
$\hat{\varphi}_{\geq 0} \leftarrow n$
**for** $r \in [m]$ **do**
   lb $= 0$, ub $= \hat{\varphi}_{\geq r-1}$             {Binary search to find $\hat{\varphi}_{\geq r}$}
   **while** ub $>$ lb $+ 1$ **do**
      mid $\leftarrow \lfloor (\text{ub} + \text{lb})/2 \rfloor$
      $v \leftarrow \Gamma^B(\hat{\varphi}_{\geq 1}, \ldots, \hat{\varphi}_{\geq r-1}, \text{mid})_r$             {See Algorithm 6}
      **if** $v \geq \varphi_{\geq r}^{\hat{\boldsymbol{n}}^{2,\mathrm{red}}}$ **then**
         ub $\leftarrow$ mid
      **else**
         lb $\leftarrow$ mid
   $v_{\text{ub}} \leftarrow \Gamma^B(\hat{\varphi}_{\geq 1}, \ldots, \hat{\varphi}_{\geq r-1}, \text{ub})_r$
   $v_{\text{lb}} \leftarrow \Gamma^B(\hat{\varphi}_{\geq 1}, \ldots, \hat{\varphi}_{\geq r-1}, \text{lb})_r$
   **if** $|v_{\text{lb}} - \varphi_{\geq r}^{\hat{\boldsymbol{n}}^{2,\mathrm{red}}}| \leq |v_{\text{ub}} - \varphi_{\geq r}^{\hat{\boldsymbol{n}}^{2,\mathrm{red}}}|$ **then**
      $\hat{\varphi}_{\geq r} \leftarrow$ lb
   **else**
      $\hat{\varphi}_{\geq r} \leftarrow$ ub
Find $\hat{\boldsymbol{n}}$ that minimizes $\sum_{r=1}^m |\hat{\varphi}_{\geq r} - \varphi_{\geq r}^{\hat{\boldsymbol{n}}}| + \sum_{r=m+1}^n |\varphi_{\geq r}^{\hat{\boldsymbol{n}}^1} - \varphi_{\geq r}^{\hat{\boldsymbol{n}}}|$ using $\ell_1$-isotonic regression algorithm
**return** $\hat{\boldsymbol{n}}$

---

**Run Time Analysis.** Recall that Algorithm 1 runs in $O(D + n \log n)$ time. In binary search, we call Algorithm 6 for a total of at most $O(m \log n) = O(\log D \log n / \varepsilon)$ times; since each call runs in

$O(nr) = O(n \log D/\varepsilon)$ time, the total running time of this step is $O(n \log n (\log D/\varepsilon)^2)$. Finally, note that the last step is again an $\ell_1$-isotonic regression problem and therefore can be solved in $O(n \log n)$ time. Thus, in total the running time is $\widetilde{O}_\varepsilon(D + n)$ as desired. Note that we may assume without loss of generality that $D \le n^{O(1)}$ as otherwise we may first hash to, e.g., $B' = O(n^2)$ buckets, which results in an increase in the expected $\ell_1$-error of at most $O(1)$.

**Error Analysis.** We now prove the error guarantee of the algorithm, which is stated precisely below.

**Theorem 25.** *Suppose that $B > 3nm$. Then, we have*

$$\mathbb{E}[\|\hat{\boldsymbol{n}} - \boldsymbol{n}\|_1] \le O(\sqrt{n^2/B} \cdot \log n) + O(\sqrt{C_p B \log n}) + O(\sqrt{C_p n \log n}).$$

Note that, since taking $p = e^{-\varepsilon/4}$ makes the algorithm $\varepsilon$-DP, we get the following:

**Corollary 26.** *For all $\varepsilon > 0$, Algorithm 7 for $p = e^{-\varepsilon/4}$ and $B = 10nm$ is $\varepsilon$-DP, and achieves an expected $\ell_1$-error of $O(\sqrt{n}(\log(nD))/\varepsilon^3)$.*

The quantitative bound above matches Corollary 13 to within a factor of $(\log n)^{1/4}/\varepsilon^{0.5}$ (recall that we can assume without loss of generality that $D \le n^{O(1)}$).

To prove Theorem 25, we will need the following auxiliary lemma, whose proof is given later in this section.

**Lemma 27.** *Suppose that $B > 3nm$. Then,*

$$\sum_{r=1}^{m} |\hat{\varphi}_{\ge r} - \hat{\varphi}_{\ge r}^{\boldsymbol{n}}| \le O\left(\|\Gamma^B(\boldsymbol{n}) - \varphi_{\ge}^{\hat{\boldsymbol{n}}^{2,\mathrm{red}}}\|_1\right). \tag{10}$$

We are now ready to prove Theorem 25.

*Proof of Theorem 25.* First, the expected $\ell_1$-error can be written as

$$\mathbb{E}[\|\hat{\boldsymbol{n}} - \boldsymbol{n}\|_1] = \mathbb{E}\left[\sum_{r=1}^{n} |\varphi_{\ge r}^{\hat{\boldsymbol{n}}} - \varphi_{\ge r}^{\boldsymbol{n}}|\right] \le O\left(\mathbb{E}\left[\sum_{r=1}^{m} |\hat{\varphi}_{\ge r} - \varphi_{\ge r}^{\boldsymbol{n}}| + \sum_{r=m+1}^{n} |\varphi_{\ge r}^{\hat{\boldsymbol{n}}^1} - \varphi_{\ge r}^{\boldsymbol{n}}|\right]\right),$$

where the last step follows from the triangle inequality and our choice of $\hat{\boldsymbol{n}}$.

We now plug in Lemma 27, which yields:

$$\mathbb{E}[\|\hat{\boldsymbol{n}} - \boldsymbol{n}\|_1] \le O\left(\mathbb{E}\left[\|\Gamma^B(\boldsymbol{n}) - \varphi_{\ge}^{\hat{\boldsymbol{n}}^{2,\mathrm{red}}}\|_1\right] + \mathbb{E}\left[\sum_{r=m+1}^{n} |\varphi_{\ge r}^{\hat{\boldsymbol{n}}^1} - \varphi_{\ge r}^{\boldsymbol{n}}|\right]\right)$$

$$\overset{\text{Lemma 15, Theorem 9}}{\le} O(\sqrt{n^2/B} \cdot \log n + \sqrt{C_p B \log n}) + \mathbb{E}\left[\sum_{r=m+1}^{n} |\varphi_{\ge r}^{\hat{\boldsymbol{n}}^1} - \varphi_{\ge r}^{\boldsymbol{n}}|\right]$$

To bound the last term, we use Observation 11, which gives:

$$\mathbb{E}\left[\sum_{r=m+1}^{n} |\varphi_{\ge r}^{\hat{\boldsymbol{n}}^1} - \varphi_{\ge r}^{\boldsymbol{n}}|\right] \le \sum_{r=m+1}^{n} \sqrt{\mathrm{Var}[\varphi_{\ge r}^{\hat{\boldsymbol{n}}^1}]} \le \sum_{r=m+1}^{n} \sqrt{\kappa} \cdot \sqrt{\sum_{\ell=0}^{n} p^{|\ell-r|} \cdot \varphi_\ell^{\boldsymbol{n}}}$$

$$\le \sqrt{\kappa} \cdot \sqrt{\sum_{r=m+1}^{n} \frac{1}{r}} \cdot \sqrt{\sum_{r=m+1}^{n} r \cdot \left(\sum_{\ell=0}^{n} p^{|\ell-r|} \cdot \varphi_\ell^{\boldsymbol{n}}\right)} \quad \text{(Cauchy–Schwarz)}$$

$$\le \sqrt{\kappa} \cdot O(\sqrt{\log n}) \cdot \sqrt{\sum_{\ell=0}^{n} \varphi_\ell^{\boldsymbol{n}} \cdot \left(\sum_{r=m+1}^{n} r \cdot p^{|\ell-r|}\right)}$$

$$\le \sqrt{\kappa} \cdot O(\sqrt{\log n}) \cdot \sqrt{\left(\varphi_0^{\boldsymbol{n}} \cdot \sum_{r=m+1}^{n} r \cdot p^r\right) + \left(\sum_{\ell=1}^{n} \varphi_\ell^{\boldsymbol{n}} \cdot \left(\sum_{r=1}^{n} r \cdot p^{|\ell-r|}\right)\right)}.$$

The second term $\sum_{\ell=1}^{n} \varphi_\ell^{\boldsymbol{n}} \cdot \left( \sum_{r=1}^{n} r \cdot p^{|\ell-r|} \right)$ can be bounded in the exact same way as in Theorem 9, which gives an upper bound of $O(\sqrt{n/(1-p)^2})$. The first term $\left( \varphi_0^{\boldsymbol{n}} \cdot \sum_{r=m+1}^{n} r \cdot p^r \right)$ can be bounded as follows:

$$\varphi_0^{\boldsymbol{n}} \cdot \sum_{r=m+1}^{n} r \cdot p^r = D\left( p^{m+1} \cdot \sum_{i=0}^{n} (i+m+1)p^i \right) \quad \leq \quad D\left( \frac{p}{D^{10}} \cdot \left( \frac{m+1}{1-p} + \frac{1}{(1-p)^2} \right) \right)$$

$$\leq O\left( \frac{1}{(1-p)^2} \right).$$

Putting the four inequalities above together, we arrive at

$$\mathbb{E}[\|\hat{\boldsymbol{n}} - \boldsymbol{n}\|_1] \leq O(\sqrt{n^2/B} \cdot \log n + \sqrt{C_p B \log n}) + O(\sqrt{\kappa/(1-p)^2}) \cdot O(\sqrt{n \log n}). \qquad \square$$

*Proof of Lemma 27.* Let $\sigma := B/n$. We will prove by induction that

$$\sum_{r=1}^{\ell} |\hat{\varphi}_{\geq r} - \hat{\varphi}_{\geq r}^{\boldsymbol{n}}| \leq 3\left(1 + \frac{3}{\sigma}\right)^{\ell} \left( \sum_{r=1}^{\ell} |\Gamma^B(\boldsymbol{n})_r - \varphi_{\geq r}^{\hat{\boldsymbol{n}}^{2,\text{red}}}| \right), \qquad (11)$$

for all $\ell = 0, \ldots, m$. The base case $\ell = 0$ is trivial.

We will next prove the inductive step. Suppose that the bound (11) holds for $\ell - 1$. We will now show that it also holds for $\ell$. Recall from Lemma 20 that $\Gamma^B(\hat{\varphi}_{\geq 1}, \ldots, \hat{\varphi}_{\geq \ell})_\ell$ is increasing in $\hat{\varphi}_{\geq \ell}$ when $\hat{\varphi}_{\geq 1}, \ldots, \hat{\varphi}_{\geq \ell-1}$ are held fixed. This means that our binary search algorithm finds the optimum, which implies

$$|\Gamma^B(\hat{\varphi}_{\geq 1}, \ldots, \hat{\varphi}_{\geq \ell-1}, \hat{\varphi}_{\geq \ell})_\ell - \varphi_{\geq \ell}^{\hat{\boldsymbol{n}}^{2,\text{red}}}| \leq |\Gamma^B(\hat{\varphi}_{\geq 1}, \ldots, \hat{\varphi}_{\geq \ell-1}, \varphi_{\geq \ell}^{\boldsymbol{n}})_\ell - \varphi_{\geq \ell}^{\hat{\boldsymbol{n}}^{2,\text{red}}}|. \qquad (12)$$

Applying the third inequality in Lemma 20 repeatedly, we have

$$|\Gamma^B(\varphi_{\geq 1}^{\boldsymbol{n}}, \ldots, \varphi_{\geq \ell-1}^{\boldsymbol{n}}, \hat{\varphi}_{\geq \ell})_\ell - \varphi_{\geq \ell}^{\hat{\boldsymbol{n}}^{2,\text{red}}}|$$

$$(\text{Lemma 20}) \quad \leq \quad |\Gamma^B(\hat{\varphi}_{\geq 1}, \ldots, \hat{\varphi}_{\geq \ell-1}, \hat{\varphi}_{\geq \ell})_\ell - \varphi_{\geq \ell}^{\hat{\boldsymbol{n}}^{2,\text{red}}}| + \frac{\sum_{r=1}^{\ell-1} |\varphi_{\geq r}^{\boldsymbol{n}} - \hat{\varphi}_{\geq r}|}{\sigma}$$

$$(\text{From (12)}) \quad \leq \quad |\Gamma^B(\hat{\varphi}_{\geq 1}, \ldots, \hat{\varphi}_{\geq \ell-1}, \varphi_{\geq \ell}^{\boldsymbol{n}})_\ell - \varphi_{\geq \ell}^{\hat{\boldsymbol{n}}^{2,\text{red}}}| + \frac{\sum_{r=1}^{\ell-1} |\varphi_{\geq r}^{\boldsymbol{n}} - \hat{\varphi}_{\geq r}|}{\sigma}$$

$$(\text{Lemma 20}) \quad \leq \quad |\Gamma^B(\varphi_{\geq 1}^{\boldsymbol{n}}, \ldots, \varphi_{\geq \ell-1}^{\boldsymbol{n}}, \varphi_{\geq \ell}^{\boldsymbol{n}})_\ell - \varphi_{\geq \ell}^{\hat{\boldsymbol{n}}^{2,\text{red}}}| + \frac{2 \cdot \sum_{r=1}^{\ell-1} |\varphi_{\geq r}^{\boldsymbol{n}} - \hat{\varphi}_{\geq r}|}{\sigma}$$

$$= \quad |\Gamma^B(\boldsymbol{n})_\ell - \varphi_{\geq \ell}^{\hat{\boldsymbol{n}}^{2,\text{red}}}| + \frac{2 \cdot \sum_{r=1}^{\ell-1} |\varphi_{\geq r}^{\boldsymbol{n}} - \hat{\varphi}_{\geq r}|}{\sigma}. \qquad (13)$$

Again applying the second inequality in Lemma 20, we get

$$|\hat{\varphi}_{\geq \ell} - \varphi_{\geq \ell}^{\boldsymbol{n}}|$$

$$\leq \frac{1}{1 - 1/\sigma} \cdot |\Gamma^B(\varphi_{\geq 1}^{\boldsymbol{n}}, \ldots, \varphi_{\geq \ell-1}^{\boldsymbol{n}}, \hat{\varphi}_{\geq \ell})_\ell - \Gamma^B(\varphi_{\geq 1}^{\boldsymbol{n}}, \ldots, \varphi_{\geq \ell-1}^{\boldsymbol{n}}, \varphi_{\geq \ell}^{\boldsymbol{n}})_\ell|$$

$$\leq \frac{1}{1 - 1/\sigma} \cdot \left( |\Gamma^B(\varphi_{\geq 1}^{\boldsymbol{n}}, \ldots, \varphi_{\geq \ell-1}^{\boldsymbol{n}}, \hat{\varphi}_{\geq \ell})_\ell - \varphi_{\geq \ell}^{\hat{\boldsymbol{n}}^{2,\text{red}}}| + |\Gamma^B(\varphi_{\geq 1}^{\boldsymbol{n}}, \ldots, \varphi_{\geq \ell-1}^{\boldsymbol{n}}, \varphi_{\geq \ell}^{\boldsymbol{n}})_\ell - \varphi_{\geq \ell}^{\hat{\boldsymbol{n}}^{2,\text{red}}}| \right)$$

$$\overset{(13)}{\leq} \frac{1}{1 - 1/\sigma} \left( |\Gamma^B(\boldsymbol{n})_\ell - \varphi_{\geq \ell}^{\hat{\boldsymbol{n}}^{2,\text{red}}}| + \frac{2 \cdot \sum_{r=1}^{\ell-1} |\varphi_{\geq r}^{\boldsymbol{n}} - \hat{\varphi}_{\geq r}|}{\sigma} + |\Gamma^B(\boldsymbol{n})_\ell - \varphi_{\geq \ell}^{\hat{\boldsymbol{n}}^{2,\text{red}}}| \right)$$

$$\leq 3|\Gamma^B(\boldsymbol{n})_\ell - \varphi_{\geq \ell}^{\hat{\boldsymbol{n}}^{2,\text{red}}}| + \frac{3 \cdot \sum_{r=1}^{\ell-1} |\varphi_{\geq r}^{\boldsymbol{n}} - \hat{\varphi}_{\geq r}|}{\sigma} \qquad \left( \text{using } \frac{1}{1 - 1/\sigma} \leq \frac{3}{2} \right).$$

Therefore, we have

$$\sum_{r=1}^{\ell} |\varphi_{\geq r}^{\boldsymbol{n}} - \hat{\varphi}_{\geq r}| \leq 3|\Gamma^B(\boldsymbol{n})_\ell - \varphi_{\geq \ell}^{\hat{\boldsymbol{n}}^{2,\text{red}}}| + \left(1 + \frac{3}{\sigma}\right) \cdot \left( \sum_{r=1}^{\ell-1} |\varphi_{\geq r}^{\boldsymbol{n}} - \hat{\varphi}_{\geq r}| \right).$$

Plugging in the inductive hypothesis, we can conclude that (12) also holds for $\ell$.

Finally, plugging in $\ell = m$ and using the fact that $\sigma \geq 3\ell$, we arrive at the claimed bound. $\qquad \square$

# E   On Pan-Privacy

In the main body of our work, we only consider the notion of pan-privacy where, for every $t \in [n]$, the internal state of the algorithm after the $t$th step must be $\varepsilon$-DP. This can be achieved for discrete Laplace-noised histogram as follows: start with $h_j$ drawn from $\mathrm{DLap}(p)$ for $p = e^{-\varepsilon/2}$ for all $j \in [D]$. Then, at each step, increment the corresponding entry $h_j$.

While this algorithm suffices for our more relaxed notion, it does not satisfy the original notion of pan-privacy as defined in [25], which requires that, for every $t \in [n]$, both the internal state of the algorithm after the $t$th step *and* the final output must be $\varepsilon$-DP. A possible adaptation of the above algorithm to satisfy this notion of pan-privacy is to also add a noise drawn from $\mathrm{DLap}(p)$ to each entry of the histogram after the last element in the stream (before computing the final output). It is simple to see that this satisfies $\varepsilon$-DP in the more restricted notion when we set $p = e^{-\varepsilon/4}$.

Unfortunately, this adaptation does *not* result in a discrete Laplace-noised histogram. Instead, the final histogram is noised by two i.i.d. discrete Laplace random variables (one from the initialization, and one from the final step). Due to this, we also have to adapt our estimation algorithm. Specifically, in Algorithm 1, we replace $f$ by[9] $f * g$ where $g$ is defined by

$$g(m) = \begin{cases} \frac{1+p^2}{(1-p)^2} & \text{if } m = 0, \\ -\frac{p}{(1-p)^2} & \text{if } m = -1 \text{ or } m = 1, \\ 0 & \text{if } m < -1 \text{ or } m > 1. \end{cases}$$

It is not hard to check that this results in an unbiased estimator, and an analogue of Theorem 9 can be proved but with $C_p = O(1/(1-p)^9)$. This in turn results in a worse error of $O(\sqrt{(n+D)\log n}/\varepsilon^{4.5})$ instead of $O(\sqrt{(n+D)\log n}/\varepsilon^{2.5})$ for the model considered in the main body. Other algorithms can be adapted similarly, again with worse dependency of $\varepsilon$ in the error bounds.

---

[9]Here $f * g$ denotes the convolution of $f$ and $g$, i.e., $(f * g)(j) = \sum_{i \in \mathbb{Z}} f(i) \cdot g(j-i)$ for all $j \in \mathbb{Z}$.