# OpenReview forum: "Anonymized Histograms in Intermediate Privacy Models"
_NeurIPS.cc/2022/Conference — NeurIPS 2022 Accept_

### Official Review · Reviewer_214Q · 2022-07-07

**Rating:** 6
**Confidence:** 2
**Soundness:** 3 good
**Presentation:** 3 good
**Contribution:** 2 fair

**Summary:**

This paper considers the release of anonymized histograms under differential privacy. In this setting, $n$ users each have some integer from the domain $[D]$. A histogram of the data is the number of users with each integer in the domain. An anonymized histogram is the set of frequencies in the histogram without the corresponding domain labels. This paper first shows that given a noisy histogram procured by adding Laplacian noise with scale parameter $\frac{1}{\varepsilon}$, there exists an algorithm that outputs an anonymized histogram with expected $\ell_1$ and $\ell_2^2$ error $O_{\varepsilon}(\sqrt{n})$. This subroutine can then be used to achieve private anonymized histograms as well as private symmetric function property testers in the pan-private and shuffle models.

**Questions:**

Are there linear time algorithms for the post-processing steps in Algorithms 1 and 2?

**Strengths And Weaknesses:**

Strengths: The main subroutine is nice and simple. Given an input (discretized) noisy histogram, the algorithm essentially builds a noisy cumulative mass function and then outputs the best fit integral histogram. The analysis is more intricate but also reasonably intuitive. To decrease the domain size from $D$ to $O(n)$, the algorithm hashes the universe into buckets of size $B=O(n)$. It then remains to bound the collision error, which intuitively should not be too high because many values of the domain must have frequency zero, while the remaining collisions can be charged to the $\ell_1$ error.

Weakness: I think the main weakness of the paper is that the post-processing algorithms do not have runtime analysis. Specifically, the runtime to output the best fit integral histogram is not analyzed and I think the efficiency of these post-processing algorithms can make a large difference in the practicality of this paper. Perhaps the authors can address this concern in the rebuttal phase.

Post-rebuttal update: I'm pleased to see details for a near-linear time post-processing steps for Algorithm 1 and 2. Though I have not gotten a chance to verify correctness, it seems my main concern is a non-issue, so I have increased my score from a 4 to a 6.

---

> ### Author Response · Authors · 2022-08-02
> **Post-processing**
>
> Indeed there are nearly-linear time algorithms for post-processing in Algorithms 1 and 2. We explain this in more detail below, and will include the discussion in the revision.
>
> **Algorithm 1:** First, we note that the computation of $\hat{\phi}\_{\geq r}$ can be done in linear time. Specifically, while the sum on the third line has as many as D terms, each term is one of the four values in equation (1), depending on whether $h’\_j$ is less than $r - 1$, equal to $r - 1$, equal to $r$, or greater than $r$. So we can compute this by first sorting the values of $h’\_j$ (in time $O(D + n)$ using bucket sort). This suffices for us to count the numbers of each value in the sum in constant time, which can then be used to compute the sum.
> As for the last step (finding $\hat{\mathbf{n}}$---or equivalently $\phi^{\hat{\mathbf{n}}}\_{\geq}$---that minimizes $\|\phi^{\hat{\mathbf{n}}}\_{\geq} - \hat{\phi}\_{\geq}\|_1$), this is exactly the $\ell\_1$-isotonic regression problem. This problem is known to be solvable in $O(n \log n)$-time; see e.g., [Rote, SOSA 2018](https://drops.dagstuhl.de/opus/volltexte/2018/10027/).
>
> **Algorithm 2:** Making this algorithm run in nearly-linear time requires a bit more work but we will sketch the argument below. The high-level idea is to solve for $\phi^{\hat{\mathbf{n}}}\_{\geq \ell}$ from small $\ell$ to larger $\ell$, by optimizing $|\Gamma^B(\hat{n})\_\ell - \phi^{\hat{n}^{red}}\_{\geq \ell}|$. (Note that $\Gamma^B(\hat{n})\_\ell$ depends only on $\phi^{\hat{\mathbf{n}}}\_{\geq 1}, \dots, \phi^{\hat{\mathbf{n}}}\_{\geq \ell}$.) Below we also need to use some approximation to make this efficient; we stress that the approximation only incurs a small amount of error and the final error remains the same as in Theorem 12.
>
>
> Here are the details. The key observation is, once we fix $\phi^{\hat{\mathbf{n}}}\_{\geq 1}, \dots, \phi^{\hat{\mathbf{n}}}\_{\geq \ell - 1}$, $\Gamma^B(\hat{n})\_\ell$ is an increasing function in $\phi^{\hat{\mathbf{n}}}\_{\geq \ell}$. For example, $\Gamma^B(\hat{n})\_1$ is equal to $B \cdot (1 - (1 - 1/B)^{\phi^{\hat{\mathbf{n}}}\_{\geq 1}})$. This means that we can solve for $\phi^{\hat{\mathbf{n}}}\_{\geq 1}$ that minimizes $|\Gamma^B(\hat{n})\_1 - \phi^{\hat{n}^{red}}\_{\geq 1}|$ by a binary search on $\phi^{\hat{\mathbf{n}}}\_{\geq 1}$. Similarly, once we fix $\phi^{\hat{\mathbf{n}}}\_{\geq 1}$, we can use binary search to find $\phi^{\hat{\mathbf{n}}}\_{\geq 2}$ that minimizes $|\Gamma^B(\hat{n})\_2 - \phi^{\hat{n}^{red}}\_{\geq 2}|$. We continue to do this until we have all of $\phi^{\hat{\mathbf{n}}}\_{\geq 1}, \dots, \phi^{\hat{\mathbf{n}}}\_{\geq n}$. This completes the description of the algorithm.
>
> To prove that this is an approximately good solution for the optimization objective, the idea is to use a similar argument as in the proof of Lemma 14 to argue that $\phi^{\hat{\mathbf{n}}}\_{\geq \ell}$ that minimizes $\Gamma^B(\hat{n})\_\ell$ is a good approximation to the optimal; the reasoning (similar to Lemma 14) is that changing $\phi^{\hat{\mathbf{n}}}\_{\geq \ell}$ by one will change $\Gamma^B(\hat{n})\_\ell$ more than changing the remaining entries of $\Gamma^B(\hat{n})$. Therefore, it is not economical to pick a non-optimal $\phi^{\hat{\mathbf{n}}}\_{\geq \ell}$. This roughly captures the main points of a nearly-linear time implementation of Algorithm 2.
>
> Finally, we would like to stress that, before our work, there was no shuffle DP / pan-private algorithm---efficient or inefficient---that achieved any non-trivial $o(n)$ error for the problem. Therefore, our results are novel in terms of the error bounds _regardless_ of the running time of the algorithm.

---

> > ### Comment · Reviewer_214Q · 2022-08-06
> > **Post-processing Runtime**
> >
> > Thanks for the brief description on the post-processing runtimes. Where in the revised version (or revised supplementary material) should I be looking for the formal description and analysis?

---

> > > ### Author Response · Authors · 2022-08-09
> > > **Fast post-processing analysis**
> > >
> > > We apologize for the confusion; we had intended to add them to the final version (not the revised version during rebuttal). Nonetheless, we’ve now added Appendix D to the (revised) supplementary material, which describes the fast $\tilde{O}_\epsilon(D + n)$-time algorithms. We note however that, due to time constraints of the rebuttal, we state & analyze a slightly modified version of Algorithm 2, which has a simpler analysis but a slightly worse error (by a factor of $O_\epsilon((\log n)^{1/4})$). We kindly request the reviewer to revisit their rating in the light of this analysis, since that was considered as the main weakness of the paper. Thank you very much in advance.

---

> > > > ### Comment · Reviewer_214Q · 2022-08-10
> > > > **Fast post-processing analysis**
> > > >
> > > > I'm pleased to see details for a near-linear time post-processing steps for Algorithm 1 and 2. Though I have not gotten a chance to verify correctness, it seems my main concern is a non-issue, so I have increased my score from a 4 to a 6.

---

### Official Review · Reviewer_c3rB · 2022-07-07

**Rating:** 7
**Confidence:** 3
**Soundness:** 3 good
**Presentation:** 4 excellent
**Contribution:** 3 good

**Summary:**


The authors study the problem of releasing anonymized (labels are not released) histograms with DP. First result is an algorithm that takes a histogram with Discrete Laplace noise, and returns an anonymized histogram with l_1 and l_2 errors of O(\sqrt(n + D)). Where n is number of records and D is the size of the histogram domain. Second main result is an algorithm that via hashing, overcomes the dependence on D in the previous bound. This algorithm can be implemented in the pan private model on in the shuffle model of DP. Last main result is an algorithm to estimate symmetric properties of distributions, entropy being an example.

**Questions:**

It seems to me that we need knowledge of the domain size D in advance. I’m curious, are there settings of practical interest where D is not known in advance and we still want to release an anonymized histogram? I’m asking because this is an important problem in the setting of DP Histograms with labels (https://proceedings.neurips.cc/paper/2019/file/b139e104214a08ae3f2ebcce149cdf6e-Paper.pdf)

**Strengths And Weaknesses:**

Strengths: The paper is written clearly, it is easy to follow, literature review is sufficient to give the reader context. Algorithms and bounds are novel and improve upon state of the art.
Weakness: Seeing the algorithms being applied in a real world problem or a simulation comparing to previous work would have been cool to motivate the reader. However, this is a theory paper and there is limited space so, although I think it would improve the paper,  I don’t think it is a big deal.

---

> ### Author Response · Authors · 2022-08-02
> **Size of $D$**
>
> Our algorithms work as long as we have access to a random hash function from $D$ to $B$ hash values. This essentially means that we need a sufficiently long random string. When we have an upper bound on $D$, this can be easily done. On the other hand, without knowing any upper bound on $D$ at all, this seems challenging as we have to sample the hash function “on the go”. This seems especially hard in the shuffle DP setting where each user has to do this in a consistent manner. We will add a discussion on this in the revision.

---

### Official Review · Reviewer_82i7 · 2022-07-10

**Rating:** 7
**Confidence:** 2
**Soundness:** 3 good
**Presentation:** 4 excellent
**Contribution:** 3 good

**Summary:**

This paper studies private anonymized histograms in shuffle DP and pan privacy models. Under $\ell_1$ loss, the proposed algorithm achieves $\tilde{O}(\sqrt{n}/\varepsilon^{2.5})$ error, which nearly matches the optimal bound in central DP model in terms of $n$. This paper also applies the proposed algorithm on shannon entropy estimation for discrete distributions.

**Questions:**

1. Under the central DP model and $\ell_2^2$ loss, the proposed algorithm is worse than Hay et al [33] (This paper has $O(\sqrt{n}/\varepsilon^3.5)$). Both algorithms use a Laplacian noised histogram and a post-processing step. I am wondering if there is any connection and differences.

2. I am not very familiar with pan privacy and the shuffled DP model. I am wondering if there is any lower bound technique that translates central DP bounds into these settings. What would break if you apply prior works for central DP under pan privacy and shuffled DP model?

**Limitations:**

This paper is theoretical and does not have a direct negative societal impact.

**Strengths And Weaknesses:**

Strengths: 1. The proposed algorithm is simple. It is based on random hashing and laplacian mechanism, which can be implemented in pan privacy and shuffled DP models.
2. This paper is well written. The structure is easy to follow.

Weaknesses: 1. It is not clear if the dependence on $\varepsilon$ is tight. Compared to the lower bound in central DP, there is a $1/\varepsilon^{1.5}$ gap.
2. Time complexity is not discussed here.

---

> ### Author Response · Authors · 2022-08-02
> **Clarifications & Comparison**
>
> **Dependence on $\epsilon$:** Yes, it is not tight; closing this gap is an interesting open question which will be emphasized in the conclusions.
>
> **Comparison to Hay et al.:** The post-processing method of Hay et al. requires the histogram bucket values to be sorted (e.g., in increasing order) *before noise addition*. This _cannot_ be done in the shuffle DP or pan-private setting. By giving a novel post-processing method, our work shows that—perhaps surprisingly!---such sorting is not required to achieve $\tilde{O}\_\epsilon(\sqrt{n})$.
>
> **Regarding central DP vs shuffle/pan-private DP:** Central DP is the most relaxed privacy notion; therefore, any lower bounds in central DP also apply to shuffle and pan-private DP models. On the other hand, as explained above for Hay et al.’s algorithm, some algorithms that work in the central DP cannot be easily translated to shuffle/pan-private DP settings.

---

> > ### Comment · Reviewer_82i7 · 2022-08-10
> > **Thank you for your response.**
> >
> > I have read the other reviewers' comments and the authors' responses to the time complexity analysis. Thanks to the authors for their feedback. All of my questions are addressed.

---

### Official Review · Reviewer_EABc · 2022-07-15

**Rating:** 6
**Confidence:** 4
**Soundness:** 3 good
**Presentation:** 3 good
**Contribution:** 3 good

**Summary:**

This paper studies the problem of releasing anonymized histograms subject to DP in various privacy models.    Privatizing anonymized histograms has been considered before, but only in the global privacy model.  This paper considers the pan privacy and shuffle DP models.  Their results show that the anonymized histograms can be approximated up to error that matches the special case problem of Count-Distinct in both the shuffle and pan privacy models.

**Questions:**

Why not consider continuous Gaussian noise for the $\ell_2$ error metric?  This can be applied in the shuffle model.  If noise must be discrete for other reasons, perhaps the Skellem distribution could also be used.

Should the $\ell_2$ error be squared?  Just want to make sure that the informal statements of theorems are supposed to be $\ell_2^2$ or $\ell_2$.

**Strengths And Weaknesses:**

Strengths:
- Anonymized histograms seem to be growing in popularity as an important problem to research.  The paper presents nice results in various privacy models.
- The introduction presents the previous work very clearly, giving a nice history of releasing anonymous histograms subject to privacy.
- The post processing function to get back an anonymous histogram from a privatized cumulative prevalence is a nice result.  In particular, prior work did not need to consider noise when the cumulative prevalence was privatized before converting back to an anonymous histogram.  This post processing function and its analysis are the main contribution of this work.
- I like that this work addressed the issue of large domains, although it does look like an "off the shelf" approach with a random hash.
- Estimating Symmetric Properties of Discrete Distributions is a nice application of the results in the paper.

Weaknesses:
- The results seem to follow from prior work on privatizing count-distinct.  Although the anonymized histogram problem is more general, the techniques in the analysis seem to be largely the same.  The main observation is that for count distinct, we need to compute the number of items that have true count larger than 0, which can then be generalized to number of items that have true count larger than r, which will result in a histogram with sensitivity of 1.  Hence the results boil down to the same analysis as before except applying it to a sensitivity 1 histogram rather than a single count.
- The general approach largely follows previous work, using Laplace mechanism and estimating the cumulative prevalence to then get an anonymous histogram.
- I would have liked to have seen more detail in Section 4, such as how hash functions are selected randomly.  Theorem 12 seems a bit hand wavy.


Minor nits:
- In the introduction, it states that $n^{\ell}$ is the $\ell$th largest element among the $h_j$'s, but should this be the *count* of the $\ell$th largest element for anonymous histograms?
- There are too many footnotes in the beginning, which should be included in the main body of the paper.

---

> ### Author Response · Authors · 2022-08-02
> **Contributions, Post-Processing, Gaussian Noise, and Error**
>
> **Main contribution**: As we indicated in the paper (lines 53–54), our _main_ contribution is the post-processing step, which is drastically different from previous works.  While the post-processing step itself is not too complicated, proving that it works (lines 104–106) is the core technical challenge. (Indeed, the main contribution of some previous works on DP anonymous histogram, e.g., Hay et al., is also to analyze different post-processing methods.)
>
> **Post-processing:** Furthermore, our post-processing is _fundamentally_ different from those in previous works (e.g., Hay et al.). At a high-level, their post-processing methods only work in the central DP setting and not in the shuffle DP/pan-private settings. At a more technical level, those works require the histogram bucket values to be sorted (e.g., in increasing order) *before noise addition*; however, this cannot be done in the shuffle DP or pan-private setting. By giving a novel post-processing method, our work shows that—perhaps surprisingly!—such a sorting is not required to achieve $\tilde{O}\_\epsilon(\sqrt{n})$.
>
> **Hash functions:** We will add more details to Section 4 regarding the choice of hash functions. (Note that, as is common, each hash function value is selected independently at random from $[B]$.)  We will also make Theorem 12 more precise, e.g., by recalling the definition of $C\_p$ from Theorem 9.
>
> **Notation**: Indeed we mean to say $n^{(\ell)}$  is the $\ell$th largest element among the $h\_j$'s.  Suppose the input is <1, 2, 1, 3, 2, 2, 4>.  The histogram is <2, 3, 1, 1> for the domain [4] and the anonymized histogram is <1, 1, 2, 3>.  In our notation, we say $n^{(1)} = 3, n^{(2)} = 2, n^{(3)} = 1$, and $n^{(4)} = 1$.  We will rewrite the text to make it less confusing (perhaps by giving an example).
> We apologize for the excessive number of footnotes (which was done to less distract the flow); we will revise to incorporate them into the main text.
>
> **On using Gaussian noise:** We can confirm that the noise does *not* have to be discrete. However, the reason we choose discrete Laplace noise is that it provides a pure-privacy (i.e., $\delta = 0$) guarantee for the pan-private setting and, even in the shuffle DP setting, it provides an error that only depends on $\epsilon$ (i.e., $\delta$ can be made arbitrarily small without increasing the error). These are in contrast to Gaussian noise which requires $\delta > 0$ even for pan-privacy and has an error that grows when $\delta$ decreases. Finally, we would like to stress that the discrete Laplace noise already allows us to achieve nearly optimal $\tilde{O}\_\epsilon(\sqrt{n})$ bound so it is unlikely that using Gaussian noise will provide a significant improvement in terms of the error.
>
> **$\ell_2^2$ error**: We confirm that the error is $\ell\_2^2$ as stated in the informal statements of the theorems as well as in the Appendix. We will emphasize this in the text.

---

> > ### Comment · Reviewer_EABc · 2022-08-07
> > **Addressing Author Feedback**
> >
> > Thanks to the authors for their feedback.  My main concerns seem to mostly just due to notation/typos, which are easily fixed.  I think the paper should highlight more the contribution of the post-processor and how this is not possible to use in the other privacy models.  Further, another reviewer also pointed out that the run time of the post processor should be addressed.  I will keep my score the same as the submission seems solid but could be better presented.

---

### Meta-Review · Area_Chair_jVH3 · 2022-08-25

**Recommendation:** Accept
**Confidence:** Certain

**Metareview:**

This paper considers the release of anonymized histograms under differential privacy, and presents new (and simple) algorithms for the shuffle model and the pan-private model of differential privacy. The reviewers all agree that the problem and the results are interesting, and support accepting the paper.

**Award:**

No

---

### Decision · Program_Chairs · 2022-09-14

Accept